# ARM: Adaptive Reasoning Model

**Siye Wu**♠    **Jian Xie**♡*    **Yikai Zhang**♠ **Aili Chen**♠ **Kai Zhang**♡ **Yu Su**♡ **Yanghua Xiao**♠♣†

♠Shanghai Key Laboratory of Data Science,
College of Computer Science and Artificial Intelligence, Fudan University
♡The Ohio State University ♣Shanghai Academy of AI for Science
siyewu24@m.fudan.edu.cn, xie.1741@osu.edu, shawyh@fudan.edu.cn

Project Page: https://team-arm.github.io/arm

## Abstract

While large reasoning models demonstrate strong performance on complex tasks, they lack the ability to adjust reasoning token usage based on task difficulty. This often leads to the "overthinking" problem—excessive and unnecessary reasoning—which, although potentially mitigated by human intervention to control the token budget, still fundamentally contradicts the goal of achieving fully autonomous AI. In this work, we propose ***Adaptive Reasoning Model*** (ARM), a reasoning model capable of adaptively selecting appropriate reasoning formats based on the task at hand. These formats include three efficient ones—*Direct Answer*, *Short CoT*, and *Code*—as well as a more elaborate format, *Long CoT*. To train ARM, we introduce ***Ada-GRPO***, an adaptation of Group Relative Policy Optimization (GRPO), which addresses the format collapse issue in traditional GRPO. Ada-GRPO enables ARM to achieve high token efficiency, reducing tokens by an average of $\sim 30\%$, and up to $\sim 70\%$, while maintaining performance comparable to the model that relies solely on *Long CoT*. Furthermore, not only does it improve inference efficiency through reduced token generation, but it also brings a $\sim 2\times$ speedup in training. In addition to the default *Adaptive Mode*, ARM supports two additional reasoning modes: 1) *Instruction-Guided Mode*, which allows users to explicitly specify the reasoning format via special tokens—ideal when the appropriate format is known for a batch of tasks. 2) *Consensus-Guided Mode*, which aggregates the outputs of the three efficient formats and resorts to *Long CoT* in case of disagreement, prioritizing performance with higher token usage.

## 1 Introduction

The emergence of large reasoning models (LRMs) such as OpenAI-o1 [19] and DeepSeek-R1 [11] has led to unprecedented breakthroughs in problem-solving capabilities through test-time scaling [3; 57]. These models are designed to solve tasks using Long Chain-of-Thought (Long CoT), generating more tokens to achieve better performance. However, because they are primarily trained on tasks requiring intensive reasoning, LRMs tend to apply Long CoT uniformly across all tasks, resulting in the so-called "overthinking" problem [4; 42]. This issue refers to the excessive use of tokens for reasoning, which yields no performance gains and may introduce noise that misleads the model [52; 7].

While some efforts aim to reduce token usage in LRMs, they often rely on clear estimations of the token budget per task [1; 46] or require specialized, length-constrained model training [16]. In reality, such estimations are not always accurate, and a more desirable solution is for models to adaptively

---

*Project lead. Part of the work was done at Fudan University.
†Corresponding author.

39th Conference on Neural Information Processing Systems (NeurIPS 2025).

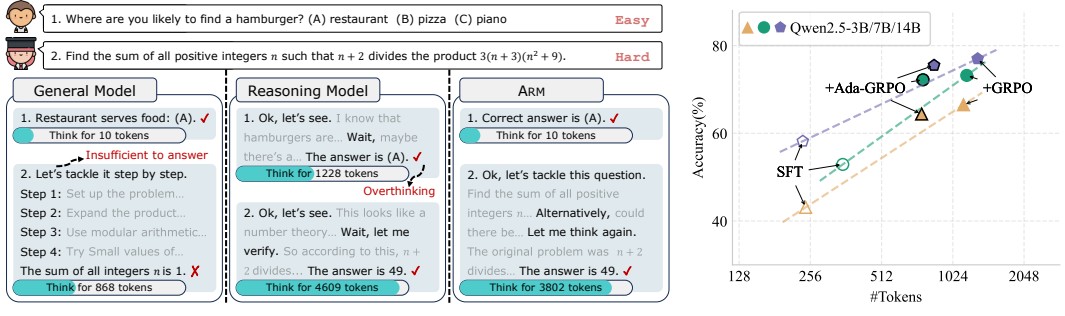

(a) Comparison of model reasoning behaviors on easy and hard tasks.     (b) Accuracy vs. Token Cost.

Figure 1: **(a)** Comparison of reasoning behaviors across different models on easy and hard tasks. The General Model fails on harder tasks without elaborate reasoning. The Reasoning Model applies Long CoT across all tasks, causing the "overthinking" phenomenon. In contrast, our proposed ARM adapts its reasoning formats based on task difficulty, answering easy questions efficiently while adopting Long CoT for hard tasks. **(b)** Accuracy versus token cost for Qwen2.5 under different training strategies. "SFT", "+GRPO", and "+Ada-GRPO" refer to models trained with SFT, SFT+GRPO, and SFT+Ada-GRPO, respectively. "+Ada-GRPO" consistently outperforms the expected trade-off line between "SFT" and "+GRPO," demonstrating ARM's superior effectiveness-efficiency balance.

control their token usage based on task complexity without human intervention. For example, for the first easy question in Figure 1a, answering directly is the ideal choice, whereas for the second, hard question, the use of Long CoT is precisely what is needed. Thus, Long CoT is not a "silver bullet"; selecting an appropriate reasoning format is essential to balance efficiency and effectiveness.

In this work, we propose ***Adaptive Reasoning Model*** (ARM), a reasoning model capable of adaptively selecting reasoning formats based on task difficulty, balancing both performance and computational efficiency. ARM supports four reasoning formats: three efficient ones—*Direct Answer*, *Short CoT*, and *Code*—and one elaborate format, *Long CoT*. In addition to its adaptive selection mechanism (*Adaptive Mode*), ARM also supports an *Instruction-Guided Mode*, which allows explicit control over the reasoning format via special tokens, and a *Consensus-Guided Mode*, which aggregates the outputs of the three efficient formats and resorts to *Long CoT* in case of disagreement.

To train ARM, we adopt a two-stage training framework. In Stage 1, we apply supervised fine-tuning (SFT) to equip the language model with a foundational understanding of four reasoning formats. In Stage 2, we introduce Ada-GRPO, an adaptation of Group Relative Policy Optimization (GRPO) [38], which encourages efficient format selection while preserving accuracy as the primary objective. Ada-GRPO is designed to address two key issues: *1)* The uniform distribution of reasoning formats regardless of task difficulty observed during the SFT stage; *2)* The format collapse problem in GRPO, where *Long CoT* gradually dominates as training progresses, leading to the diminished use of other, more efficient formats. Extensive evaluations show that ARM trained with Ada-GRPO achieves comparable performance while using $\sim 30\%$ fewer tokens than GRPO (as shown in Figure 1b), across both in-domain and out-of-domain tasks in commonsense, mathematical, and symbolic reasoning. Furthermore, by leveraging the three more efficient reasoning formats in the roll-out stage, Ada-GRPO achieves approximately a $2\times$ training speedup compared to GRPO.

Our additional analysis further reveals that: *1)* Adaptive Mode achieves a superior balance between effectiveness and token efficiency by adaptively selecting suitable reasoning formats, while Instruction-Guided Mode performs well when the specified format is suitable for the task, and Consensus-Guided Mode prioritizes performance at the cost of higher token usage. *2)* The choice of backbone model has limited impact on ARM's performance when using base or instruction-tuned models, which yield similar results; however, using the DeepSeek-R1-Distill backbone improves performance on hard tasks due to the advanced reasoning capability distilled from the strong teacher DeepSeek-R1, but leads to worse performance on easy tasks despite increased token cost. *3)* Length-penalty-based strategies for improving LRM efficiency suffer from performance degradation as the token budget decreases, whereas ARM maintains stable performance.

In summary, our contributions are three-fold: *1)* We propose ARM, a reasoning model that balances effectiveness and efficiency by adaptively selecting task-appropriate reasoning formats. Compared to

the model that relies solely on *Long CoT*, ARM achieves comparable performance while significantly reducing token cost, saving an average of $\sim 30\%$ and up to $\sim 70\%$. *2)* In addition to the default Adaptive Mode, ARM also supports Instruction-Guided Mode, which performs well when the reasoning format is appropriately specified, and Consensus-Guided Mode, which maximizes performance at the cost of higher token usage. *3)* We introduce Ada-GRPO, an adaptation of GRPO that addresses the format collapse problem and achieves a $\sim 2\times$ training speedup without compromising performance.

## 2 Related Work

### 2.1 Reinforcement Learning for Improving Reasoning

Reinforcement Learning (RL) has demonstrated significant potential in enhancing the problem-solving abilities of large language models (LLMs) across various domains [33; 47; 20]. Recently, Reinforcement Learning with Verifiable Rewards (RLVR) has gained substantial attention for advancing LLM capabilities [21; 11; 25], resulting in the development of large reasoning models (LRMs) [53] such as OpenAI-o1 [19] and DeepSeek-R1 [11]. Based on simple rule-based rewards, RLVR algorithms such as Group Relative Policy Optimization (GRPO) [38] enable models to use Long Chain-of-Thought (Long CoT) [56; 18]. This facilitates deep reasoning behaviors, such as searching, backtracking, and verifying through test-time scaling [3; 57]. However, these models also suffer from significant computational overhead due to extended outputs across all tasks, leading to inefficiency associated with the "overthinking" phenomenon [4; 35; 42]. Verbose and redundant outputs can obscure logical clarity and hinder the model's ability to solve problems effectively [52; 7].

### 2.2 Efficiency in Large Language Models

Recently, many studies have focused on improving reasoning efficiency in LLMs. Some prompt-guided methods [12; 13; 54; 22] explicitly instruct LLMs to generate concise reasoning outputs by controlling input properties such as task difficulty and response length. Other approaches [14; 5; 39] explore training LLMs to reason in latent space, generating the *direct answer* without the need for detailed language tokens. Several techniques have also been proposed to reduce inference costs by controlling or pruning output length, either by injecting multiple reasoning formats during the pre-training stage [41] or by applying length penalties during the RL stage [45; 2; 1; 16]. Many of these methods aim to strike a trade-off between token budget and reasoning performance by shortening output lengths, often relying on clear estimations of the token budget for each task or requiring specialized, length-constrained model training. However, in reality, such estimations are not always accurate, and what we ultimately expect is for models to adaptively regulate their token usage based on the complexity of the task at hand. Therefore, in this work, we propose a novel training framework that enables models to adaptively select suitable reasoning formats for given tasks by themselves, optimizing both performance and computational efficiency.

## 3 Method

We propose Adaptive Reasoning Model (ARM), a reasoning model designed to optimize effectiveness and efficiency by adaptively selecting reasoning formats. Specifically, ARM is trained in two stages: *1)* **Stage 1: Supervised Fine-tuning (SFT) for Reasoning Formats Understanding**: In this stage, we use 10.8K diverse questions, each annotated with solutions in four distinct reasoning formats, to fine-tune the model and build a foundational understanding of different reasoning strategies. *2)* **Stage 2: Reinforcement Learning (RL) for Encouraging Efficient Format Selection**: We adopt an adapted version of the GRPO algorithm, named Ada-GRPO, to train the model to be capable of selecting more efficient reasoning formats over solely *Long CoT*, while maintaining accuracy.

### 3.1 Stage 1: SFT for Reasoning Formats Understanding

In this stage, we leverage SFT as a cold start to introduce the model to various reasoning formats it can utilize to solve problems.[3] These formats include three efficient reasoning formats *Direct*

---

[3]In preliminary experiments, models without SFT failed to distinguish between the four reasoning formats, like producing mixed outputs, wrapping a *Short CoT* response using the special tokens intended for *Long CoT*.

*Answer*, *Short CoT*, and *Code*, as well as the elaborate reasoning format *Long CoT*. We use special tokens (e.g., **) to embrace thinking rationale. Specifically, *1) **Direct Answer**:* This format provides a direct answer without any reasoning chain, making it the most efficient in terms of token usage. *2) **Short CoT**:* This format begins with a short reasoning and then provides an answer, which has been proved effective in mathematical problems [49]. *3) **Code**:* This format adopts code-based reasoning, which has proven effective across a variety of tasks due to its structured process [50; 51; 24]. *4) **Long CoT**:* This format involves a more detailed, iterative reasoning process, thus incurs higher token usage. It is suited for tasks requiring advanced reasoning capabilities, such as self-reflection and alternative generation, where those more efficient formats fall short [31; 11; 56].

## 3.2 Stage 2: RL for Encouraging Efficient Format Selection

After SFT, the model learns to respond using various reasoning formats but lacks the ability to adaptively switch between them based on the task (see Section 4.3 for details). To address this, we propose **Ada**ptive GRPO (**Ada-GRPO**), which enables the model to dynamically select appropriate reasoning formats according to the task difficulty through a format diversity reward mechanism.

**GRPO**     In traditional GRPO [38], the model samples a group of outputs $O = \{o_1, o_2, \cdots, o_G\}$ for each question $q$, where $G$ denotes the group size. For each $o_i$, a binary reward $r_i$ is computed using a rule-based reward function that checks whether the prediction $pred$ matches the ground truth $gt$:

$$r_i = \mathbb{1}_{\texttt{passed}(gt, pred)}. \tag{1}$$

However, since traditional GRPO solely optimizes for accuracy, it leads, in our setting, to overuse of the highest-accuracy format while discouraging exploration of alternative reasoning formats. Specifically, if *Long CoT* achieves higher accuracy than other formats, models trained with GRPO tend to increasingly reinforce it, leading to an over-reliance on *Long CoT* and reduced exploration of more efficient alternatives. We refer to this phenomenon as **Format Collapse**, which ultimately hinders the model's ability to develop adaptiveness. We further analyze this in Section 4.3.

**Ada-GRPO**     We propose Ada-GRPO to address the format collapse issue. Specifically, Ada-GRPO amplifies the reward $r_i$ for less frequently sampled reasoning formats, preventing their disappearance and ensuring adequate learning. Formally, we scale the reward $r_i$ to $r'_i$ by:

$$r'_i = \alpha_i(t) \cdot r_i, \tag{2}$$

$$\alpha_i(t) = \frac{G}{F(o_i)} \cdot decay_i(t), \tag{3}$$

$$decay_i(t) = \frac{F(o_i)}{G} + 0.5 \cdot \left(1 - \frac{F(o_i)}{G}\right) \cdot \left(1 + \cos\left(\pi \cdot \frac{t}{T}\right)\right), \tag{4}$$

where $F(o_i)$ denotes the number of times the reasoning format corresponding to $o_i$ appears within its group $O$, and $t$ represents the training step. $\alpha_i(t)$ is a format diversity scaling factor that gradually decreases from $\frac{G}{F(o_i)}$ at the beginning of training ($t = 0$) to 1 at the end of training ($t = T$).

We introduce $\alpha_i(t)$ to extend GRPO into **Ada-GRPO**, enabling models to adaptively select reasoning formats. Specifically, $\alpha_i(t)$ consists of two components: *1)* **Format Diversity Scaling Factor** $\frac{G}{F(o_i)}$: To prevent premature convergence on the highest-accuracy format (i.e., format collapse to *Long CoT*), we upweight rewards for less frequent formats to encourage exploration. *2)* **Decay Factor** $decay_i(t)$: To avoid long-term misalignment caused by over-rewarding rare formats, this term gradually reduces the influence of diversity over time. For example, $\frac{G}{F(o_i)}$ might make the model favor a lower-accuracy format like *Short CoT* over *Long CoT* simply because it appears less frequently and thus receives a higher reward. While such exploration is beneficial early in training, it can hinder convergence later. The decay mechanism mitigates this by promoting diversity initially, then shifting focus to accuracy again as training progresses. Refer to Appendix A for details of the decay factor.

Then the group advantage $\hat{A}_{i,k}$ for all tokens in each output is computed based on the group of reshaped rewards $\mathbf{r}' = \{r'_1, r'_2, \cdots, r'_G\}$:

$$\hat{A}_{i,k} = \frac{r'_i - \text{mean}(\{r'_1, r'_2, \cdots, r'_G\})}{\text{std}(\{r'_1, r'_2, \cdots, r'_G\})}. \tag{5}$$

Finally, we optimize the model by maximizing the following objective (see Appendix A for details):

$$
\mathcal{J}_{\text{Ada-GRPO}}(\theta) = \mathbb{E}\left[q \sim P(Q), \{o_i\}_{i=1}^G \sim \pi_{\theta_{\text{old}}}(O|q)\right]\left[\frac{1}{\sum_{i=1}^G |o_i|}\sum_{i=1}^G\sum_{k=1}^{|o_i|}\left\{\min\left[\frac{\pi_\theta(o_{i,k}|q, o_{i,<k})}{\pi_{\theta_{\text{old}}}(o_{i,k}|q, o_{i,<k})}\hat{A}_{i,k},\right.\right.\right.
$$
$$
\left.\left.\left.\text{clip}\left(\frac{\pi_\theta(o_{i,k}|q, o_{i,<k})}{\pi_{\theta_{\text{old}}}(o_{i,k}|q, o_{i,<k})}, 1-\epsilon, 1+\epsilon\right)\hat{A}_{i,k}\right] - \beta\,\text{KL}\left[\pi_\theta \parallel \pi_{\text{ref}}\right]\right\}\right].
$$
$$(6)$$

## 4 Experiment

### 4.1 Experimental Setup

**Model**    To assess the effectiveness of our method across different model sizes, we select Qwen2.5-Base-3B/7B/14B [55] as backbone models. We further examine models of the same family but fine-tuned on different datasets, specifically the *Instruct* [55] and *DeepSeek-R1-Distill* variants [11], which exhibit varying levels of base reasoning capabilities. A detailed analysis is given in Section 5.3.

**Training Datasets**    *Stage 1:*  We use AQuA-Rat [26] as the SFT dataset, as its answers can be naturally transformed into four distinct reasoning formats. In addition to the *Direct Answer* and *Short CoT* rationales provided with the dataset, we utilize GPT-4o [30] and DeepSeek-R1 [11] to supplement the *Code* and *Long CoT* rationales, respectively. To ensure the quality of the generated rationales, we filter out those that lead to incorrect answers, resulting in a training set containing 3.0K multiple-choice and 7.8K open-form questions, each with four reasoning formats. Appendix B provides further details on the generation and filtering process. *Stage 2:*  To prevent data leakage, we employ three additional datasets exclusively for the RL stage.[4] These datasets cover a range of difficulty levels, from relatively simple commonsense reasoning tasks to more complex mathematical reasoning tasks, including CommonsenseQA (CSQA) [44], GSM8K [6], and MATH [15], collectively comprising 19.8K verifiable question-answer pairs. Please refer to Appendix C for details of the datasets we use and Appendix D for implementation details.

**Baselines**    In addition to backbone models, we compare ARM with models trained using alternative algorithms that may enable adaptive reasoning capabilities. Specifically, **Qwen2.5SFT** refers to the backbone models trained on the AQuA-Rat dataset used in Stage 1. In this setting, we explore whether language models can master adaptive reasoning through a straightforward SFT strategy. For **Qwen2.5SFT+GRPO**, we examine whether SFT models, further trained with GRPO, can better understand different reasoning formats and whether this approach empowers them to select appropriate reasoning formats based on rule-based rewards.

### 4.2 Evaluation

**Evaluation Datasets**    To assess the models' reasoning capabilities, we select a range of evaluation datasets, including both in-domain and out-of-domain samples. These datasets span commonsense, mathematical, and symbolic reasoning tasks. For commonsense reasoning, we include CommonsenseQA (CSQA)[44] and OpenBookQA (OBQA)[29], which are easier tasks based on intuitive knowledge. For mathematical reasoning, we utilize SVAMP [34], GSM8K [6], MATH [15], and AIME'25 [9] to assess models' ability to solve complex mathematical problems that require advanced reasoning and strict logical thinking. For symbolic reasoning, we turn to Big-Bench-Hard (BBH) [43], a benchmark for evaluating models' structured reasoning ability to manipulate symbols according to formal rules. For further analysis, we group the evaluation datasets into three difficulty levels: commonsense tasks as *easy*; mathematical and symbolic tasks as *medium*; and AIME'25 as *hard* given its competition-level difficulty.

**Inference**    During inference, we set the temperature to 0.7 and top-p to 1.0. For all evaluation datasets, we use accuracy as the metric. In addition to pass@1, to reduce bias and uncertainty associated with single generation outputs and to enhance the robustness of the results [57], we further use majority@k (maj@k), which measures the correctness of the majority vote from $k$ independently

---

[4]In preliminary experiments, we observed that using the same training data in both stages causes the model to recite answers rather than reasoning during the RL stage, resulting in poor generalization.

Table 1: Performance of various models across evaluation datasets. "#Tokens" refers to the token cost for each model on each dataset. For each model, $k = 1$ corresponds to pass@1, and $k = 8$ corresponds to maj@8. When $k = 8$, the token cost is averaged over a single output to facilitate clear comparison. "†" denotes in-domain tasks, while "‡" denotes out-of-domain tasks. "$\Delta$" represents the difference between ARM and Qwen2.5$_{\text{SFT+GRPO}}$, calculated by subtracting the accuracy of Qwen2.5$_{\text{SFT+GRPO}}$ from that of ARM, with the token usage expressed as the ratio of tokens saved by ARM compared to Qwen2.5$_{\text{SFT+GRPO}}$, with all settings based on $k = 8$ to ensure a stable comparison.

| Models | k | Accuracy (↑) | | | | | | | | #Tokens (↓) | | | | | | | |
| | | Easy | | Medium | | | | Hard | Avg. | Easy | | Medium | | | | Hard | Avg. |
| | | CSQA† | OBQA‡ | GSM8K† | MATH† | SVAMP‡ | BBH‡ | AIME'25‡ | | CSQA† | OBQA‡ | GSM8K† | MATH† | SVAMP‡ | BBH‡ | AIME'25‡ | |
|---|---|---|---|---|---|---|---|---|---|---|---|---|---|---|---|---|---|
| GPT-4o | 1 | 85.9 | 94.2 | 95.9 | 75.9 | 91.3 | 84.7 | 10.0 | 76.8 | 192 | 165 | 287 | 663 | 156 | 278 | 984 | 389 |
| o1-preview | 1 | 85.5 | 95.6 | 94.2 | 92.6 | 92.7 | 91.8 | 40.0 | 84.6 | 573 | 492 | 456 | 1863 | 489 | 940 | 7919 | 1819 |
| o4-mini-high | 1 | 84.7 | 96.0 | 96.9 | 97.7 | 94.0 | 92.2 | 96.7 | 94.0 | 502 | 289 | 339 | 1332 | 301 | 755 | 9850 | 1910 |
| DeepSeek-V3 | 1 | 82.4 | 96.0 | 96.5 | 91.8 | 93.7 | 85.8 | 36.7 | 83.3 | 231 | 213 | 236 | 887 | 160 | 400 | 2992 | 732 |
| DeepSeek-R1 | 1 | 83.3 | 94.8 | 96.4 | 97.1 | 96.0 | 85.0 | 70.0 | 88.9 | 918 | 736 | 664 | 2339 | 589 | 1030 | 9609 | 2270 |
| DS-R1-Distill-1.5B | 1 | 47.6 | 48.6 | 79.4 | 84.6 | 86.7 | 53.5 | 20.0 | 60.1 | 987 | 1540 | 841 | 3875 | 606 | 3005 | 13118 | 3425 |
| DS-R1-Distill-7B | 1 | 64.9 | 77.4 | 90.0 | 93.6 | 90.3 | 72.1 | 40.0 | 75.5 | 792 | 928 | 574 | 3093 | 315 | 1448 | 12427 | 2797 |
| DS-R1-Distill-14B | 1 | 80.6 | 93.2 | 94.0 | 95.5 | 92.7 | 80.4 | 50.0 | 83.8 | 816 | 750 | 825 | 2682 | 726 | 1292 | 11004 | 2585 |
| DS-R1-Distill-32B | 1 | 83.2 | 94.6 | 93.5 | 93.0 | 92.0 | 86.3 | 56.7 | 85.6 | 674 | 698 | 438 | 2161 | 283 | 999 | 11276 | 2361 |
| Qwen2.5-3B | 1 | 66.5 | 65.8 | 66.9 | 37.7 | 71.3 | 38.4 | 0 | 49.5 | 97 | 120 | 150 | 419 | 76 | 232 | 1393 | 355 |
| | 8 | 75.5 | 77.4 | 80.9 | 50.8 | 83.7 | 47.1 | 0 | 59.3 | 96 | 100 | 149 | 424 | 85 | 240 | 1544 | 377 |
| Qwen2.5-3B$_{\text{SFT}}$ | 1 | 72.8 | 72.4 | 35.7 | 20.9 | 62.3 | 37.4 | 0 | 43.1 | 99 | 108 | 145 | 229 | 126 | 311 | 694 | 245 |
| | 8 | 75.5 | 77.4 | 56.0 | 27.6 | 74.7 | 43.5 | 0 | 50.7 | 97 | 103 | 132 | 231 | 108 | 309 | 537 | 217 |
| Qwen2.5-3B$_{\text{SFT+GRPO}}$ | 1 | 79.7 | 79.0 | 88.7 | 66.6 | 92.0 | 52.6 | 6.7 | 66.5 | 425 | 501 | 788 | 1586 | 630 | 994 | 3027 | 1136 |
| | 8 | 80.3 | 80.0 | 91.4 | 74.0 | 94.7 | 56.2 | 6.7 | 69.0 | 429 | 506 | 802 | 1590 | 638 | 996 | 3247 | 1172 |
| ARM-3B | 1 | 79.8 | 78.0 | 83.8 | 62.9 | 89.7 | 50.0 | 6.7 | 64.4 | 118 | 156 | 346 | 1013 | 264 | 436 | 2958 | 756 |
| | 8 | 80.1 | 78.0 | 90.8 | 72.8 | 95.0 | 53.8 | 6.7 | 68.2 | 123 | 169 | 359 | 1036 | 246 | 430 | 3083 | 778 |
| Δ | | -0.2 | -2.0 | -0.6 | -1.2 | +0.3 | -2.4 | 0 | -0.8 | -71.3% | -66.6% | -55.2% | -34.8% | -61.4% | -56.8% | -5.1% | -33.6% |
| Qwen2.5-7B | 1 | 76.7 | 78.6 | 81.6 | 50.1 | 81.0 | 51.7 | 3.3 | 60.4 | 64 | 83 | 156 | 376 | 99 | 182 | 767 | 247 |
| | 8 | 82.0 | 86.4 | 89.9 | 64.7 | 89.7 | 62.0 | 3.3 | 68.3 | 66 | 74 | 156 | 370 | 92 | 183 | 881 | 260 |
| Qwen2.5-7B$_{\text{SFT}}$ | 1 | 80.8 | 81.2 | 54.4 | 30.4 | 76.0 | 48.2 | 0 | 53.0 | 136 | 150 | 184 | 348 | 126 | 245 | 1239 | 347 |
| | 8 | 83.9 | 84.6 | 79.4 | 42.4 | 88.0 | 56.0 | 0 | 62.0 | 141 | 137 | 185 | 361 | 141 | 274 | 1023 | 323 |
| Qwen2.5-7B$_{\text{SFT+GRPO}}$ | 1 | 83.1 | 82.2 | 92.8 | 79.4 | 93.7 | 64.3 | 16.7 | 73.2 | 491 | 651 | 739 | 1410 | 587 | 1133 | 3196 | 1173 |
| | 8 | 83.7 | 84.6 | 94.8 | 84.9 | 95.3 | 69.3 | 20.0 | 76.1 | 496 | 625 | 745 | 1415 | 586 | 1135 | 3145 | 1164 |
| ARM-7B | 1 | 86.1 | 84.4 | 89.2 | 73.9 | 92.0 | 61.4 | 16.7 | 72.0 | 136 | 159 | 305 | 889 | 218 | 401 | 3253 | 766 |
| | 8 | 85.7 | 85.8 | 94.8 | 82.6 | 95.3 | 67.9 | 20.0 | 75.9 | 134 | 154 | 297 | 893 | 218 | 413 | 3392 | 786 |
| Δ | | +2.0 | +1.2 | -1.1 | -2.3 | 0 | -1.4 | 0 | -0.2 | -73.0% | -75.4% | -60.1% | -36.9% | -62.8% | -63.6% | +7.9% | -32.5% |
| Qwen2.5-14B | 1 | 79.9 | 83.8 | 84.9 | 52.7 | 84.7 | 56.8 | 3.3 | 63.7 | 56 | 60 | 132 | 335 | 77 | 139 | 611 | 201 |
| | 8 | 83.8 | 90.2 | 92.3 | 68.4 | 91.7 | 67.4 | 3.3 | 71.0 | 55 | 60 | 131 | 325 | 81 | 131 | 735 | 217 |
| Qwen2.5-14B$_{\text{SFT}}$ | 1 | 81.8 | 88.0 | 62.6 | 37.4 | 84.0 | 53.5 | 0 | 58.2 | 155 | 140 | 161 | 276 | 152 | 254 | 527 | 238 |
| | 8 | 85.0 | 91.4 | 86.4 | 48.8 | 91.7 | 64.4 | 3.3 | 67.3 | 149 | 141 | 165 | 288 | 140 | 247 | 493 | 232 |
| Qwen2.5-14B$_{\text{SFT+GRPO}}$ | 1 | 85.4 | 93.0 | 94.8 | 81.7 | 93.7 | 70.5 | 20.0 | 77.0 | 558 | 531 | 693 | 1805 | 565 | 945 | 4031 | 1304 |
| | 8 | 85.8 | 94.2 | 96.1 | 87.1 | 95.3 | 77.0 | 20.0 | 79.4 | 552 | 537 | 696 | 1810 | 565 | 943 | 3723 | 1261 |
| ARM-14B | 1 | 85.3 | 91.8 | 92.5 | 79.1 | 93.3 | 66.6 | 20.0 | 75.5 | 146 | 128 | 294 | 903 | 212 | 420 | 3871 | 853 |
| | 8 | 85.6 | 91.8 | 96.3 | 86.4 | 95.7 | 72.1 | 23.3 | 78.7 | 145 | 134 | 293 | 910 | 189 | 415 | 3996 | 869 |
| Δ | | -0.2 | -2.4 | +0.2 | -0.7 | +0.4 | -4.9 | +3.3 | -0.7 | -73.7% | -75.0% | -57.9% | -49.7% | -66.5% | -56.0% | +7.3% | -31.1% |

sampled outputs. For inference on the three backbone models, we use an example with a short-cot-based answer within the prompt to guide the model toward specific answer formats while preserving its original reasoning capabilities as much as possible.

## 4.3 Main Results

Alongside our baselines, we include several state-of-the-art general models, including GPT-4o [30] and DeepSeek-V3 [27], as well as reasoning models o1-preview [31], o4-mini-high [32], and DeepSeek-R1 [11], along with several DeepSeek-R1-Distill-Qwen (DS-R1-Distill) models ranging from 1.5B to 32B [11]. We report our results in Table 1, and we have the following findings:

**Current reasoning models struggle with the "overthinking" problem, with smaller distilled models being more affected.** We observe that all current reasoning models consume more than 500 tokens on easy commonsense tasks but do not always achieve corresponding improvements. For example, although DeepSeek-R1 and DS-R1-Distill-7B use nearly 4× and 10× more tokens than their backbone models, DeepSeek-V3 and Qwen2.5-7B, they do not show significant improvement and even experience performance degradation, highlighting the "overthinking" problem. Additionally, we find that when comparing different sizes of DS-R1-Distill, smaller models often require more tokens while delivering worse performance.

**SFT only teaches models about formats, yet does not teach how to choose the appropriate formats based on the task.** We observe that SFT models, across three sizes, show improvement on easy commonsense tasks but experience performance drops on medium and hard tasks. To investigate the cause, we conduct a deeper analysis of the reasoning formats selected during inference. Figure 2 visualizes how models allocate the four reasoning formats across three difficulty levels. Specifically, we find that for models trained with SFT, their outputs are distributed almost uniformly across the reasoning formats, with the majority in *Direct Answer* and the least in *Long CoT*, regardless of task difficulty. As shown in Figure 2, the inappropriate selection of *Direct Answer*, which yields extremely low accuracy (35.2%) on medium tasks and significantly hinders the model's reasoning capabilities,

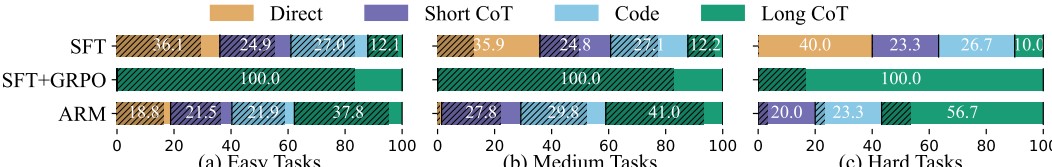

Figure 2: Format distribution by task difficulty with Qwen2.5-7B. The hatched areas indicate the percentage of correct answers that were generated using the selected reasoning format.

Table 2: Accuracy (Acc.) and token usage (Tok.) for the three reasoning modes supported by ARM-7B. In the Consensus-Guided Mode, the percentage of *Long CoT* usage indicates how often the model resorts to *Long CoT* when simpler reasoning formats fail to reach a consensus.

| ARM-7B | Easy | | | | Medium | | | | | | | | Hard | | Avg. | |
|---|---|---|---|---|---|---|---|---|---|---|---|---|---|---|---|---|
| | CSQA† | | OBQA‡ | | GSM8K† | | MATH† | | SVAMP‡ | | BBH‡ | | AIME'25‡ | | | |
| | Acc. | Tok. | Acc. | Tok. | Acc. | Tok. | Acc. | Tok. | Acc. | Tok. | Acc. | Tok. | Acc. | Tok. | Acc. | Tok. |
| Adaptive | 86.1 | 136 | 84.4 | 159 | 89.2 | 305 | 73.9 | 889 | 92.0 | 218 | 61.4 | 401 | 16.7 | 3253 | 72.0 | 766 |
| Inst$_{Direct}$ | 84.1 | 10 | 81.8 | 10 | 22.9 | 11 | 23.1 | 13 | 67.0 | 11 | 44.7 | 21 | 0 | 12 | 46.2 | 13 |
| Inst$_{Short\ CoT}$ | 81.3 | 33 | 77.4 | 35 | 85.0 | 124 | 70.9 | 633 | 86.7 | 66 | 49.7 | 101 | 10.0 | 2010 | 65.9 | 428 |
| Inst$_{Code}$ | 84.4 | 140 | 81.6 | 147 | 84.2 | 285 | 65.9 | 559 | 88.3 | 182 | 57.9 | 344 | 10.0 | 1821 | 67.5 | 497 |
| Inst$_{Long\ CoT}$ | 84.0 | 259 | 87.4 | 294 | 91.8 | 426 | 77.2 | 1220 | 94.3 | 340 | 66.9 | 660 | 20.0 | 4130 | 74.5 | 1047 |
| Consensus | 85.8 | 228 | 87.0 | 260 | 92.9 | 777 | 78.4 | 2281 | 95.7 | 433 | 66.4 | 1039 | 20.0 | 7973 | 75.2 | 1856 |
| *Long CoT* Usage | 12.9% | | 21.4% | | 79.8% | | 79.2% | | 36.3% | | 56.3% | | 100% | | 55.1% | |

finally leads to a decline in overall performance. This suggests that while SFT teaches models various formats, it fails to help them adaptively select appropriate ones based on the task, leading to an inability to choose more advanced formats as problem complexity increases.

**GRPO does improve reasoning capabilities, but it tends to rely on *Long CoT* to solve all tasks.** We observe that models trained with GRPO achieve significant improvements across all tasks, yet the token cost remains substantial, especially for the two easier tasks. Further analysis reveals that *Long CoT* is predominantly used in the inference stage, as shown in Figure 2. This behavior stems from the nature of GRPO (i.e., format collapse discussed in Section 3.2), where models converge to the format with the highest accuracy (i.e., *Long CoT*) early in training (∼10 steps in our experiment). As a result, GRPO also fails to teach models how to select a more efficient reasoning format based on the task. We provide more details of format collapse in Appendix E.

**ARM is able to adaptively select reasoning formats based on task difficulty, while achieving comparable accuracy across all tasks compared to GRPO and using significantly fewer tokens.** As shown in Table 1, across three different model sizes, all ARMs experience an average performance drop of less than 1% compared to models trained with GRPO, yet they save more than 30% of the tokens. Specifically, ARM demonstrates a clear advantage on easy tasks, saving over 70% of tokens while maintaining comparable accuracy. This advantage extends to medium tasks as well. For the more challenging AIME'25 task, ARM adapts to the task difficulty by increasingly selecting *Long CoT*, thereby avoiding performance degradation on harder tasks, with ARM-14B even surpassing its counterpart Qwen2.5-14B$_{SFT+GRPO}$. Figure 2 further confirms that ARM is able to gradually adopt more advanced reasoning formats and discards simpler ones as task difficulty increases. Moreover, as shown in Figure 1b, the line connecting "SFT" and "+GRPO" illustrates the expected trade-off, while "+Ada-GRPO" consistently lies above it, indicating a better balance between effectiveness and efficiency of ARM. Additionally, ARM-7B achieves comparable performance to DS-R1-Distill-7B while using only 27.8% of the tokens on average. For a broader view of generalization across additional benchmarks, please refer to Appendix F.

### 4.4 Reasoning Mode Switching

ARM is capable of autonomously selecting appropriate reasoning formats (Adaptive Mode), while also supporting explicit guidance to reason in specified formats (Instruction-Guided Mode) or through consensus between different reasoning formats (Consensus-Guided Mode). Specifically, *1)* **Adaptive Mode:** In this mode, ARM autonomously selects the reasoning format for each task, which is also the

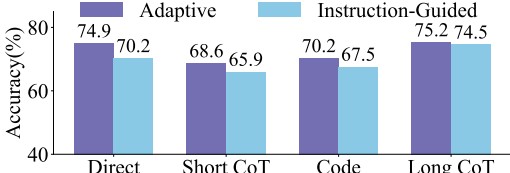

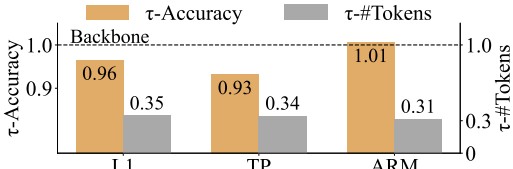

Figure 3: Accuracy comparison between ARM's **Adaptive** and **Instruction-Guided** modes. The figure shows average accuracy across evaluation datasets, with *Direct Answer* applied only to commonsense and symbolic tasks, as it does not appear in mathematical tasks in Adaptive mode.

Figure 4: Relative accuracy and token usage of different models compared to their backbone models on CSQA. "L1" denotes L1-Exact [1], and "TP" denotes THINKPRUNE [16]. "$\tau$-Accuracy" and "$\tau$-#Tokens" are reported relative to each model's backbone after RL training.

default reasoning mode if not specified in this paper. *2)* **Instruction-Guided Mode:** In this mode, a specific token (e.g., *<Long CoT>*) is provided as the first input, forcing ARM to reason in the specified format. *3)* **Consensus-Guided Mode:** In this mode, ARM first generates answers using the three simpler reasoning formats (i.e., *Direct Answer*, *Short CoT*, and *Code*) and checks for consensus among them. If all formats agree, the consensus answer is adopted as the final result. Otherwise, ARM defaults to *Long CoT* for the final answer, treating the task as sufficiently complex.

To evaluate the performance and effectiveness of the proposed reasoning modes, we conduct experiments across various evaluation datasets. Table 2 presents the results for ARM-7B. Specifically: *1)* **Adaptive Mode strikes a superior balance between high accuracy and efficient token usage across all datasets, demonstrating its ability to adaptively select the reasoning formats.** *2)* **Instruction-Guided Mode offers a clear advantage when the assigned reasoning format is appropriate.** For example, *Direct Answer* is sufficient for commonsense tasks, while *Code*, due to its structured nature, performs better on symbolic reasoning tasks compared to *Direct Answer* and *Short CoT*. Furthermore, Inst$_{Long\ CoT}$ achieves better performance (74.5%) than the same-sized model trained on GRPO (73.2% in Table 1). This demonstrates that Ada-GRPO does not hinder the model's *Long CoT* reasoning capabilities. We further validate this by analyzing the reflective words used by ARM-7B and Qwen2.5-7B$_{SFT+GRPO}$ in Appendix G. *3)* **Consensus-Guided Mode, on the other hand, is performance-oriented, requiring more tokens to achieve better performance.** This mode leverages consensus across multiple formats to mitigate bias and uncertainty present in any single format, offering greater reliability, particularly for reasoning tasks that demand advanced cognitive capabilities, where simpler formats may fall short. This is evidenced by the fact that *Long CoT* is less likely to be used for easy tasks, but is highly likely to be selected for medium tasks and even used 100% of the time for the most difficult AIME'25 task.

## 5 Analysis

### 5.1 Effectiveness of Adaptive Format Selection

To verify that ARM's format selection indeed adapts to the task at hand rather than relying on random selection, we compare ARM's **Adaptive Mode** with **Instruction-Guided Mode**. In Instruction-Guided Mode, the reasoning format is fixed and manually specified, providing a strong baseline to test whether adaptive selection offers real benefits over using a uniform format across tasks. We report the accuracy of both modes in Figure 3. We observe that the accuracy of the reasoning formats selected in Adaptive Mode is higher than that in Instruction-Guided Mode. Specifically, Adaptive Mode improves accuracy by 4.7% on *Direct Answer*, by 2.7% on both *Short CoT* and *Code*, and even yields a slight improvement on *Long CoT*. These results confirm that ARM is not randomly switching formats but is instead learning to select an appropriate one for each task. Further ablation results on reasoning formats are presented in Appendix H.

### 5.2 Comparison of Ada-GRPO and GRPO

We find that, compared to GRPO, ARM trained with Ada-GRPO achieves comparable performance on the evaluation dataset while achieving approximately a $\sim 2\times$ speedup in training time. To understand the source of this efficiency, we compare the training dynamics of Ada-GRPO and GRPO across

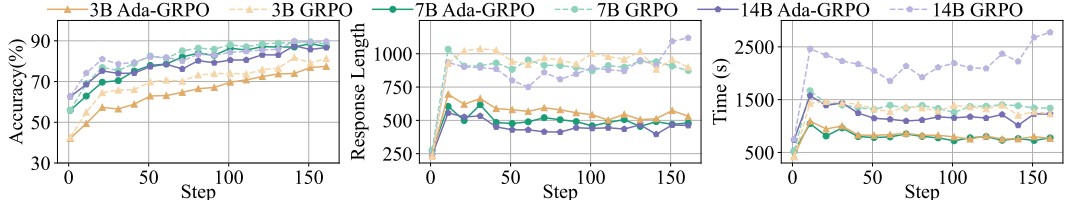

Figure 5: Performance on the training set across different model sizes trained with Ada-GRPO and GRPO. Except for the implementation of the algorithm, all hyperparameters are kept the same.

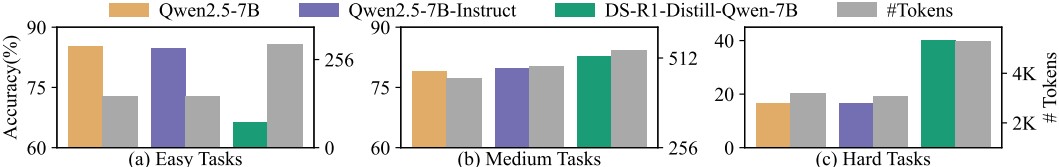

Figure 6: ARMs' performance across different backbones. Base and instruction-tuned models perform similarly, while DS-R1-Distill improves on medium and hard tasks but struggles on easy ones.

different model sizes, focusing on accuracy, response length, and training time, as shown in Figure 5. The results highlight the following advantages of Ada-GRPO: *1)* **Comparable Accuracy.** Although Ada-GRPO initially lags behind GRPO in accuracy due to suboptimal reasoning format selection in the early training steps, both methods converge to similar final accuracy across all model sizes. This demonstrates that Ada-GRPO does not compromise final performance. *2)* **Half Response Length.** While GRPO uses *Long CoT* uniformly across all tasks, Ada-GRPO adaptively selects reasoning formats based on task difficulty. Due to the length efficiency of *Direct Answer*, *Short CoT*, and *Code*, Ada-GRPO ultimately reduces the average response length to roughly half that of GRPO. *3)* **Half Training Time Cost.** Since the majority of training time is spent on response generation during the roll-out stage, reducing response length directly translates into lower time cost. As a result, Ada-GRPO achieves approximately a $\sim 2\times$ speedup compared to GRPO. Overall, Ada-GRPO maintains strong performance while significantly reducing computational overhead, underscoring its efficiency and reliability for training.

### 5.3 Comparison of Backbone Models

Beyond the base model, we further analyze the impact of different backbone models, including instruction-tuned and DS-R1-Distill variants. Figure 6 reports accuracy and token usage across *easy*, *medium*, and *hard* tasks. We observe that base and instruction-tuned models have a highly similar performance. This suggests that RL effectively bridges the gap left by instruction tuning, enabling base models to achieve comparable performance, consistent with findings from previous work [20]. In contrast, the DS-R1-Distill variant performs notably better on medium and hard tasks, benefiting from distilled knowledge from the stronger DeepSeek-R1 model, though at the expense of increased token cost. However, it performs significantly worse on easy tasks, even with excessive token usage, resulting from the overthinking phenomenon. Additional discussion and case studies on the overthinking phenomenon are presented in Appendix I, and a complementary analysis of LLaMA-based backbones is included in Appendix J.

### 5.4 Comparison of ARM and Length-Penalty-Based Strategies

To examine whether previously proposed length-penalty-based strategies—proven effective in complex reasoning—remain effective for easier tasks, we evaluate two representative methods, L1 [1] and THINKPRUNE [16], on the CSQA dataset. Since both methods are based on the DS-R1-Distill model, we ensure a fair comparison by also evaluating the version of ARM trained on the same backbone. We report the relative accuracy and token usage of all three models compared to their respective backbone models in Figure 4. When using the minimum allowed lengths specified in the official settings of L1 and THINKPRUNE, both methods exhibit performance drops. In contrast, ARM maintains strong performance while using relatively fewer tokens, demonstrating its ability to balance reasoning efficiency and effectiveness. Please see Appendix K for further discussion and details.

# 6   Conclusion

In this work, we propose *Adaptive Reasoning Model* (ARM), which adaptively selects reasoning formats based on task difficulty. ARM is trained with Ada-GRPO, a GRPO variant that addresses format collapse via a format diversity reward and achieves a $\sim 2\times$ training speedup. Experiments show that ARM maintains performance comparable to the GRPO-trained model relying solely on *Long CoT*, while significantly improving token efficiency. Beyond the default Adaptive Mode, ARM also supports Instruction-Guided Mode, which excels when the format is appropriately specified, and Consensus-Guided Mode, which maximizes performance at higher token usage. By adopting the adaptive reasoning format selection strategy, ARM effectively mitigates the overthinking problem and offers a novel, efficient approach to reducing unnecessary reasoning overhead.

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

# Appendix

## A Details of Ada-GRPO

### A.1 Training Objective

Following GRPO [38], given a query $q$ and a set of responses $O = \{o_1, o_2, \ldots, o_G\}$ sampled from the old policy $\pi_{\text{old}}$, we optimize the policy model $\pi$ using the Ada-GRPO objective:

$$
\mathcal{J}_{\text{Ada-GRPO}}(\theta) = \mathbb{E}\left[q \sim P(Q), \{o_i\}_{i=1}^G \sim \pi_{\theta_{\text{old}}}(O|q)\right] \left[\frac{1}{\sum_{i=1}^G |o_i|} \sum_{i=1}^G \sum_{k=1}^{|o_i|} \left\{ \min\left[\frac{\pi_\theta(o_{i,k}|q, o_{i,<k})}{\pi_{\theta_{\text{old}}}(o_{i,k}|q, o_{i,<k})}\hat{A}_{i,k}, \right.\right.\right.
$$
$$
\left.\left.\left. \text{clip}\left(\frac{\pi_\theta(o_{i,k}|q, o_{i,<k})}{\pi_{\theta_{\text{old}}}(o_{i,k}|q, o_{i,<k})}, 1-\epsilon, 1+\epsilon\right)\hat{A}_{i,k}\right] - \beta\, \text{KL}\left[\pi_\theta \parallel \pi_{\text{ref}}\right]\right\}\right],
$$
$$
\tag{7}
$$

where $\pi_{\text{ref}}$ denotes the reference model, and the KL divergence term KL serves as a constraint to prevent the updated policy from deviating excessively from the reference. The advantage estimate $\hat{A}_{i,k}$ is computed based on a group of rewards $\{r'_1, r'_2, \cdots, r'_G\}$ associated with the responses in $O$, as defined in equation 5.

### A.2 Decay Factor

In Ada-GRPO, the decay factor $decay_i(t)$ is introduced to regulate the influence of the format diversity scaling factor during training. Without decay, the model may continue to overly reward less frequent reasoning formats even after sufficient exploration, misaligning with our objective. To evaluate the effectiveness of the decay mechanism, we track the test set performance across three in-domain datasets (CSQA, GSM8K, and MATH) using checkpoints saved every 25 training steps for models trained with and without decay. As shown in Figure 7, models trained without decay exhibit larger performance fluctuations in test accuracy, indicating unstable exploration. In contrast, the decay mechanism stabilizes training, resulting in smoother and more consistent improvements in accuracy during the middle and later training stages.

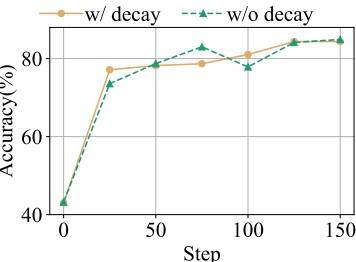

Figure 7: Test set accuracy with and without the decay mechanism.

## B Details of Processing SFT Dataset

### B.1 Prompt List

We use gpt-4o-2024-11-20 to generate *Code* reasoning rationales. Following previous work [50], we ask the model to return the output as a dictionary containing all intermediate and final outputs, which is beneficial for emulating the generated program's execution.

```
For the following questions and answers, generate a function that
solves the question. The function should return a dictionary with the
field 'answer': <answer>, as well as the values for intermediate
decisions. Ensure that both the function and its call are wrapped in <
CODE>..., and that the emulation of its execution is wrapped in
 <OUTPUT>...</OUTPUT>.

{examples}

Question:
{question}
Answer:
{rational}
#### {ground_truth}
```

Table 3: Dataset in each training stage.

| Dataset | Answer Format | Size |
|---------|---------------|------|
| **Stage 1: Supervised Finetuning** | | |
| AQuA-Rat | Multiple-Choice | 3.0K |
| | Open-Form | 7.8K |
| | | 10.8K |
| **Stage 2: Reinforcement Learning** | | |
| CSQA | Multiple-Choice | 4.9K |
| GSM8K | Open-Form | 7.4K |
| MATH | Open-Form | 7.5K |
| | | 19.8K |

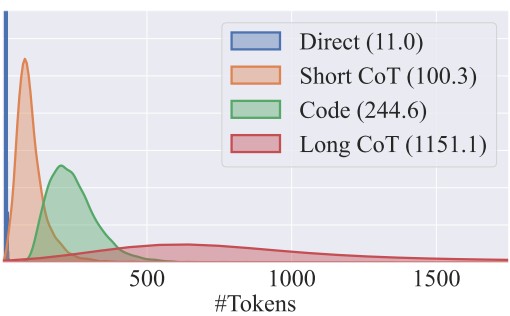

Figure 8: Token count distribution across reasoning formats in the SFT dataset AQuA-Rat, with brackets indicating average counts.

We use Deepseek-R1 to generate *Long CoT* rationales.

```
Put the answer in format "<ANSWER>...</ANSWER>".

{question}
```

### B.2 Filter Out Rationales

For *Code* rationales, we utilize a Python interpreter to execute each generated code snippet. We apply the following filters: *1)* execution failure, *2)* missing *answer* key, *3)* inconsistencies between intermediate steps and execution results, and *4)* mismatches between the predicted and ground-truth answers. For *Long CoT* rationales, we filter out those with incorrect answers. Token count distribution across reasoning formats in the SFT dataset can be seen in Figure 8.

## C   Examples in Training

The question-answer format and size of the datasets in each training stage can be found in Table 3.

### C.1   Examples in SFT Stage

We provide four reasoning formats for the SFT dataset AQuA-Rat [26], including *Direct Answer*, *Short CoT*, *Code*, *Long CoT*. Here is an example.

```
Question:
You collect baseball cards. Suppose you start out with 15. Maria takes
 half of one more than the number of baseball cards you have. Since
you're nice, you give Peter 1 baseball card. Since his father makes
baseball cards, Paul decides to triple your baseball cards. How many
baseball cards do you have at the end?

Direct Answer:
18

Code:
def calculate_baseball_cards():
    # Initial number of baseball cards
    initial_cards = 15

    # Maria takes half of one more than the number of baseball cards
    maria_takes = (initial_cards + 1) / 2
    cards_after_maria = initial_cards - maria_takes

    # Peter takes 1 baseball card
```

```
        cards_after_peter = cards_after_maria - 1

        # Paul triples the number of baseball cards
        final_cards = cards_after_peter * 3

        return {
            'maria_takes': maria_takes,
            'cards_after_maria': cards_after_maria,
            'cards_after_peter': cards_after_peter,
            'answer': final_cards
        }

>>> calculate_baseball_cards()

output = {
    'maria_takes': 8.0,
    'cards_after_maria': 7.0,
    'cards_after_peter': 6.0,
    'answer': 18.0
}
```

Short CoT:
Start with 15 baseball cards. Maria takes half of one more than the number of baseball cards you have. So Maria takes half of 15 + 1 which is 8, so you're left with 15 - 8 = 7. Peter takes 1 baseball card from you: 7 - 1 = 6 baseball cards. Paul triples the number of baseball cards you have: 6 * 3 = 18 baseball cards. So you have 18 at the end. The answer is 18.

Long CoT:
Okay, let me try to work through this problem step by step. So, the question is about collecting baseball cards, and there are a series of steps that happen. Let me read it again to make sure I understand each part.

Alright, starting with 15 cards. Then Maria takes half of one more than the number I have. Wait, that phrasing is a bit tricky. So Maria takes half of (one more than my current number of cards). Let me parse that. If I have N cards, Maria takes (N +1)/2? Hmm. Let me double-check. The problem says: "Maria takes half of one more than the number of baseball cards you have." So, the expression would be ( (number you have + 1 ) ) / 2. Right. So that's (current cards +1) divided by two. Then after that, I give Peter 1 card. Then Paul triples my cards. So, let's break it down step by step.

Starting with 15 cards. Let's note each step:

1. Initial cards: 15.

2. Maria takes half of one more than what I have. So, first, calculate one more than my current count: 15 +1 =16. Then she takes half of that. So 16 /2 =8. So Maria takes 8 cards. Wait, but when she takes them, does that mean my total decreases by 8? Yes, I think so. Because she's taking them from me. So subtract 8 from my current count.

So after Maria takes them: 15 -8 =7. Now I have 7 cards left.

Wait a minute, wait. Wait, when Maria takes half of one more than the number I have, which is (15+1)/2=8. So she's taking 8 cards. So yes, subtract 8 from my current 15. That leaves me with 7.

Then, the next step: since I'm nice, I give Peter 1 baseball card. So subtract 1 more. 7 -1 =6.

```
Then Paul decides to triple my baseball cards. So multiply my current
count by 3. 6 *3 =18.

So the answer should be 18. Let me go through the steps again to check
.

Original: 15.

After Maria: (15 +1)/2 =8. So 15-8=7.

After giving Peter 1: 7-1=6.

After Paul triples: 6*3=18.

Yes, that seems right. Did I misinterpret any step? Let's double-check
 each part.

"Maria takes half of one more than the number of baseball cards you
have." So when Maria takes, it's (my current cards +1)/2. So when I
have 15, that's (15+1)/2=8. She takes 8, so I have 7 left. Correct.

Then giving Peter 1: 7-1=6. Then Paul triples to 18. Yep. That seems
correct. So the final answer is 18.
```

## C.2 Examples in RL Stage

The RL dataset includes CSQA [44], GSM8K [6], and MATH [15], with examples provided below.

```
<CommonsenseQA>
What is the base of a vase called?
(A)transistor  (B)vessel  (C)box  (D)pie  (E)seabase
ground truth: B

<GSM8K>
Natalia sold clips to 48 of her friends in April, and then she sold
half as many clips in May. How many clips did Natalia sell altogether
in April and May?
ground truth: 72

<MATH>
Rationalize the denominator: $\frac{1}{\sqrt{2}-1}$. Express your
answer in simplest form.
ground truth: $\boxed{\sqrt{2}+1}$
```

# D Implementation Details

Our training is performed using 8 NVIDIA A800 GPUs. The following settings are also applied to other baselines for fair comparisons.

## D.1 Stage 1: SFT

We utilize the open-source training framework LLAMAFACTORY [58] to perform SFT. The training is conducted with a batch size of 128 and a learning rate of 2e-4. We adopt a cosine learning rate scheduler with a 10% warm-up period over 6 epochs. To enhance training efficiency, we employ parameter-efficient training via Low-rank adaptation (LoRA) [17] and DeepSpeed training with the ZeRO-3 optimization stage [36]. As a validation set, we sample 10% of the training data and keep the checkpoint with the lowest perplexity on the validation set for testing and the second stage.

Table 4: Effect of training with AIME-only data on reasoning format distribution.

| 7B Models | CSQA | | OBQA | |
|---|---|---|---|---|
| | *Long CoT* | Other Formats | *Long CoT* | Other Formats |
| Training with AIME only | 79.4% | 20.6% | 83.0% | 17.0% |

Table 5: Comparison of accuracy and token usage between different training recipes.

| 7B Models | CSQA | | OBQA | |
|---|---|---|---|---|
| | Acc. | Tok. | Acc. | Tok. |
| Training with AIME only | 78.6 | 401 | 82.2 | 426 |
| ARM Recipe | 86.1 | 136 | 84.4 | 159 |

## D.2 Stage 2: RL

We utilize the open-source training framework VeRL [40] to perform RL. During training, we use a batch size of 1024 and generate 8 rollouts per prompt ($G = 8$), with a maximum rollout length of 4096 tokens. The model is trained with a mini-batch size of 180, a KL loss coefficient of 1e-3, and a total of 9 training epochs. The default sampling temperature is set to 1.0.

## E   Effect of Training Data Bias on Format Collapse

Since training data may implicitly favor certain reasoning formats, it is important to examine whether such bias can induce format collapse. To this end, we analyze an ARM trained solely on the AIME dataset (1983-2024), which primarily favors *Long CoT* solutions due to its competition-level complexity. We present the result in Table 4. We evaluate this model on two simpler tasks—CSQA and OBQA—and observe that the model overwhelmingly selects *Long CoT* (∼80%) even when simpler formats would suffice. This confirms that training on a biased dataset can indeed lead to over-reliance on a single reasoning format.

To mitigate this, ARM is trained on a diverse mixture of datasets across a wide range of difficulties. As shown in Table 5, the full ARM recipe achieves both higher accuracy and significantly reduced token usage on the tasks, demonstrating that our approach effectively prevents format collapse and encourages adaptive reasoning behavior across domains.

## F   Generalization to Additional Benchmarks

As shown in Section 4.2, our existing evaluation spans across in- and out-of-domain commonsense, mathematical, and symbolic reasoning. To further investigate the generalizability of ARM, we extend the evaluation to two additional benchmarks: *1)* **GPQA** [37], a challenging QA dataset designed to test compositional reasoning, and *2)* **StrategyQA** [8], a benchmark of yes/no questions that require implicit multi-hop reasoning.

As shown in Table 6, ARM achieves comparable accuracy to the GRPO baseline while significantly reducing token cost by an average of 50%, and up to 65% in StrategyQA, consistent with our main findings discussed in Section 4.3. This demonstrates that ARM's adaptive reasoning behavior generalizes well to more tasks.

## G   Details of Reflective Words

To evaluate models' *Long CoT* reasoning capabilities, we focus on their use of specific *reflective words* that signal backtracking and verifying during the reasoning process. Following prior work [28], we consider a curated list of 17 reflective words: [*"re-check"*, *"re-evaluate"*, *"re-examine"*, *"re-think"*, *"recheck"*, *"reevaluate"*, *"reexamine"*, *"reevaluation"*, *"rethink"*, *"check again"*, *"think again"*, *"try again"*, *"verify"*, *"wait"*, *"yet"*, *"double-check"*, *"double check"*]. We adopt two evaluation metrics:

Table 6: Comparison between the GRPO baseline and ARM on GPQA and StrategyQA benchmarks.

| Models | GPQA-Main | | GPQA-Diamond | | StrategyQA | |
|---|---|---|---|---|---|---|
| | Acc. | Tok. | Acc. | Tok. | Acc. | Tok. |
| Qwen2.5-7B$_{\text{SFT+GRPO}}$ | 35.0 | 2324 | 37.4 | 2604 | 72.9 | 646 |
| ARM-7B | 34.8 | 1306 | 36.9 | 1536 | 73.8 | 229 |
| $\Delta$ | -0.2 | $-43.8\%$ | -0.5 | $-41.0\%$ | +0.9 | $-64.6\%$ |

Table 7: Definitions and results of reflection-related ratios on AIME'25.

| Ratio Name | Formula | Qwen2.5-7B$_{\text{SFT+GRPO}}$ | ARM-7B |
|---|---|---|---|
| reflection_ratio | $\frac{\mathcal{N}_{ref}}{\mathcal{N}}$ | 93.8 | 95.0 |
| correct_ratio_in_reflection_texts | $\frac{\mathcal{N}_{ref+}}{\mathcal{N}_{ref}}$ | 14.2 | 13.9 |

`reflection_ratio`, measuring the proportion of outputs containing at least one reflective word, and `correct_ratio_in_reflection_texts`, assessing the correctness within reflective outputs. The formulas for these metrics are summarized in Table 7, where $\mathcal{N}$ denotes the total number of responses, $\mathcal{N}_{ref}$ the number of responses containing reflective words, and $\mathcal{N}_{ref+}$ the number of correct reflective responses.

Given its competition-level difficulty, we conduct our analysis on AIME'25 using ARM-7B and Qwen2.5-7B$_{\text{SFT+GRPO}}$. For ARM-7B, we use the Instruction-Guided Mode (Inst$_{Long\ CoT}$) to specifically assess its *Long CoT* reasoning. The results, averaged over 8 runs, are reported in Table 7. As shown, both models exhibit a high frequency of reflective word usage, with `reflection_ratio` exceeding 93%, indicating that reflection behavior is well-integrated during *Long CoT* reasoning. The `correct_ratio_in_reflection_texts` remains comparable for both models, and relatively low due to the high complexity of the AIME'25 tasks. These results demonstrate that Ada-GRPO does not hinder the model's *Long CoT* reasoning capabilities.

## H   Ablation Study on Reasoning Formats

To further investigate whether defining other reasoning formats would help, we add an additional reasoning format: *function-calling*, implemented via code execution. Specifically, during inference, when the model selects the *Code* reasoning format, we run the generated code using an interpreter to obtain an answer. If execution fails, the model falls back to simulating the output of the code, consistent with prior work [23].

As shown in Table 8, incorporating the *function-calling* format yields improvements, suggesting that more fine-grained formats can provide benefits. However, we also note that incorporating additional reasoning formats demands extra resources to implement the pipeline. For example, parallel function calls during training can lead to high memory consumption, and decreasing the number of processes may prolong the training time. Furthermore, formats like *function-calling* would introduce additional inference-time latency (e.g., due to runtime execution). Therefore, we adopt the current four reasoning formats for their widespread use and practicality, and leave the exploration of more reasoning formats to future work.

In addition to adding reasoning formats, we further examine the effect of removing specific reasoning formats on performance and token efficiency. Specifically, during inference, we remove *Direct Answer* on CSQA, *Short CoT* on GSM8K, and *Long CoT* on AIME'25. The results are summarized in Table 9. On CSQA, removing *Direct Answer* increases token usage by +29.4% with negligible accuracy gain, showing it is crucial for efficiently handling simple tasks. In contrast, on AIME'25, removing *Long CoT* leads to a significant accuracy drop (-8.4), confirming its importance for complex reasoning. Overall, these results validate the necessity of the predefined reasoning formats in enabling adaptive reasoning.

Table 8: Accuracy across benchmarks when adding a function-calling reasoning format to ARM.

| 7B Models | Easy | | Medium | | | | Hard | Avg. |
|---|---|---|---|---|---|---|---|---|
| | CSQA | OBQA | GSM8K | MATH | SVAMP | BBH | AIME'25 | |
| vanilla ARM | 86.1 | 84.4 | 89.2 | 73.9 | 92.0 | 61.4 | 16.7 | 72.0 |
| ARM + function-calling | 86.1 | 84.5 | 90.3 | 74.3 | 92.8 | 62.1 | 16.7 | 72.4 |
| Δ | 0.0 | +0.1 | +1.1 | +0.4 | +0.8 | +0.7 | 0.0 | +0.4 |

Table 9: Effect of removing reasoning formats from ARM.

| 7B Models | CSQA | | GSM8K | | AIME'25 | |
|---|---|---|---|---|---|---|
| | Acc. | Tok. | Acc. | Tok. | Acc. | Tok. |
| vanilla ARM | 86.1 | 136 | 89.2 | 305 | 16.7 | 3253 |
| after removing | 86.2 | 176 | 89.5 | 385 | 8.3 | 2137 |
| Δ | +0.1 | +29.4% | +0.3 | +26.2% | -8.4 | −34.3% |

# I    Details of the Overthinking Phenomenon

Overthinking refers to the phenomenon where LLMs apply unnecessarily complex reasoning to simple tasks, leading to diminishing returns in performance [42]. As demonstrated in Table 1 and 2, using *Long CoT*, despite incurring higher computation costs, significantly enhances model performance on tasks requiring complex mathematical reasoning, such as MATH. However, as mentioned in Section 4.3 and 5.3, longer responses do not consistently lead to better performance for all task types. In this section, we analyze the overthinking phenomenon in depth, focusing on how overly complex reasoning formats can hurt performance when applied to certain tasks.

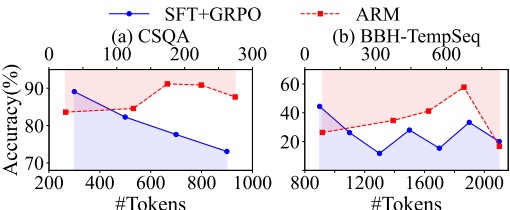

Figure 9: Overthinking in 7B model performance across two representative datasets. We remove the extreme data points and ensure that sufficient data points fall within the intervals.

## I.1    Analysis

We analyze the evaluation datasets and illustrate the "overthinking" phenomenon using two representative datasets: CSQA and TemporalSequences in BBH Benchmark. From Figure 9 (note that the horizontal axis scales differ between *SFT+GRPO* and ARM.), we observe that *SFT+GRPO*, which relies heavily on *Long CoT*, shows a significant drop in accuracy as the model generates lengthy responses. It indicates that the model starts generating excessive information that does not contribute to task resolution. In contrast, ARM achieves better performance with fewer tokens for simpler tasks. This analysis reinforces the idea that longer reasoning does not equate to better performance [7; 42], and the optimal reasoning format depends on task type. ARM provides a more flexible, task-specific solution to avoid overthinking, ultimately improving performance. Please refer to Appendix I.3 for a detailed example of how *Long CoT* errs in CSQA, while a simpler reasoning format with lower token usage successfully solves the problem.

## I.2    Data Examples

```
<BBH-Temporal Sequences>
Today, Ashley went to the bookstore. Between what times could they
have gone?
We know that: Ashley woke up at 5am. Jason saw Ashley buying lunch at
the deli from 11am to 1pm. Emily saw Ashley driving to the water park
from 1pm to 3pm. Kimberly saw Ashley stretching at a yoga studio from
3pm to 6pm.
```

```
The bookstore was closed after 6pm. Between what times could Ashley
have gone to the bookstore?
(A) 1pm to 3pm
(B) 11am to 1pm
(C) 3pm to 6pm
(D) 5am to 11am
ground truth: D
```

## I.3   An Example of Overthinking in CSQA

```
Where does a maid empty a trash can?
(A)bus stop  (B)corner  (C)hockey game  (D)motel  (E)alley

Long CoT used by +SFT+GRPO:
<LONG_COT>
Okay, let's see. The question is where a maid would empty a trash can.
 Hmm, pick one of the options given. Let's break down the terms.

First, a maid is someone who works in a hotel or other environment,
such as cleaning. Their job might involve domestic duties for a
specific kind of client, perhaps upper class, but specific job would
involve handling various items. So having cleaning duties like trash
removal.

Now, traditionally, maid service might be focused on particular areas-
probably the rooms in a hotel, apartment, or maybe an individual
client if they work as an in-home maid. So for that_RESULTS, would
probably be... Alright. Let's look at the choices:

(A) Bus stop. Seems like they shouldn't dispose of anything there. A
bus stop would have trash cans, but personally I doubt a maid would
dump trash in a bus station. That's probably not part of their
responsibilities.

(B) Corner. Hmm. A literal street corner, perhaps maybe a public
restroom? Or other outdoor spot? But where tomorrow's trash can? I can
't imagine a location in caring for trash outside normal public areas.
 Maybe mega cities and sanitation handling upgrades, but again not a
typical everyday task for a maid.

(C) Hockey Game. Hmm. Trash removal at a hockey game... Never heard of
 a maid working at a hockey game. Probably no. The grounds might be
cleaned by different personnel, like stadium employees, most likely
more specialized. Hell, playerName duplication, maybe the trash cans
outside the arena get cleaned? That probably exclude maid's work issue
. So probably can be discounted.

(D) Motel. Hmm. Maybe the version of this? Or hotels. Wait, motel is
final answer. But wait, official work responsibility. The main point
here is where professional maid service has responsibilities. For
example, in hotels-motel rooms (though maybe not modern motels, which
had more community facilities?) The specific task is they probably
need to take trash cans to the disposal area-usually outside the room
in compilations. So perhaps dump it into a proper receptacle. That
would happen in backend trash can disposal, but Solutions associated
with cleaning, not directly in guest areas.

(E) Alley. Hmm. Alley avoids touristy trash? Wait, perhaps like
backstage scenario. Or alleyways can be garbage collection areas. But
how typical? Probably the communities are serviced by regular
collection with professionals. For same reasons as airliner (like
trash in the alley). The main thing is that probably would move trash
to the nearest dumpster, possibly in an alley. Though not a public
```

```
area of the house. Though again, problem is which alley. It's not
nearly as common would be this responsibility. Probably more likely
that a maid would take the trash to a dumpster outside. So option E
probably can be considered.

Wait, not sure, but none of the options are obvious, but Zeeland E.
Because the equipment is usually moved from rooms to a centralized
dumpster. Probably near the alley. Although may not be the primary
course of action. However, this is what might be discussed. Because
depending on the structure. For example, perhaps motel rooms need to
dispose garbage in dumpster, making alley collection possible.

But options like motel and motel would probably be the right answer.
But museum? Not typical. So perhaps neither of the answers given. Wait
, maybe E.
</LONG_COT>
<ANSWER>
E
</ANSWER>

Short CoT used by ARM:
<SHORT_COT>
A maid works in a motel. A maid may empty a trash can in a motel.
The answer is D
</SHORT_COT>
<ANSWER>
D
</ANSWER>
```

## J  Experiment on LLaMA Baseline

Based on the superior performance of the Qwen model and its inherent adaptability to reinforcement learning [55; 48], we adopt Qwen as the backbone model in our experiments. To examine the generality of our method, we further train models on LLaMA-3.2-3B [10] and report the results in Table 10. Consistent with our main findings, ARM achieves comparable performance to the GRPO baseline while using fewer tokens across diverse task domains and complexity levels, demonstrating that our method can generalize to different backbone models.

We also note that token reduction is less pronounced on LLaMA than on Qwen (e.g., 15.7% vs. 55.2% on GSM8K). Upon closer analysis, we find this is caused by repetitive outputs on occasion produced by the LLaMA-based model—a phenomenon also observed in prior work [48]—which may lead to longer response lengths. This discrepancy may stem from differences in model architecture or pretraining data, and we leave further investigation to future work.

Table 10: Comparison of the GRPO baseline and ARM on LLaMA-3.2-3B [10] across CSQA, GSM8K, and AIME'25. ARM achieves similar accuracy while reducing token usage.

| LLaMA-3.2-3B | | CSQA | | GSM8K | | AIME'25 | |
|---|---|---|---|---|---|---|---|
| | k | Acc. | Tok. | Acc. | Tok. | Acc. | Tok. |
| GRPO Baseline | 1 | 76.4 | 347 | 87.5 | 677 | 3.3 | 4616 |
| | 8 | 76.8 | 350 | 90.3 | 662 | 3.3 | 4375 |
| ARM | 1 | 76.2 | 158 | 86.1 | 546 | 3.3 | 3534 |
| | 8 | 76.5 | 162 | 89.8 | 558 | 3.3 | 3713 |
| Δ | | -0.3 | -53.7% | -0.5 | -15.7% | 0 | -15.1% |

Table 11: Performance of L1-Exact under different specified token budgets across benchmarks. "Spec." indicates the user-specified reasoning budget in tokens.

| Spec. | CSQA Acc. | CSQA Tok. | AIME Acc. | AIME Tok. | MATH Acc. | MATH Tok. | AMC Acc. | AMC Tok. | olympiad_bench Acc. | olympiad_bench Tok. |
|---|---|---|---|---|---|---|---|---|---|---|
| 512 | 45.8 | 328 | 3.3 | 623 | 71.0 | 590 | 47.0 | 641 | 31.7 | 608 |
| 1024 | 46.6 | 589 | 6.7 | 1291 | 77.2 | 1182 | 45.8 | 1283 | 37.2 | 1184 |
| 2048 | 46.0 | 2004 | 13.3 | 1935 | 79.6 | 1751 | 55.4 | 1950 | 39.7 | 1813 |
| 3600 | 46.1 | 4747 | 26.7 | 3696 | 81.8 | 3478 | 72.3 | 3525 | 43.7 | 3460 |

Table 12: Comparison between L1 and ARM across multiple benchmarks.

| 7B Models | k | Easy CSQA Acc. | Easy CSQA Tok. | Easy OBQA Acc. | Easy OBQA Tok. | Medium GSM8K Acc. | Medium GSM8K Tok. | Medium MATH Acc. | Medium MATH Tok. | Medium SVAMP Acc. | Medium SVAMP Tok. | Medium BBH Acc. | Medium BBH Tok. | Hard AIME'25 Acc. | Hard AIME'25 Tok. | Avg. Acc. | Avg. Tok. |
|---|---|---|---|---|---|---|---|---|---|---|---|---|---|---|---|---|---|
| L1 | 1 | 62.4 | 232 | 69.2 | 341 | 89.8 | 273 | 85.9 | 943 | 89.7 | 231 | 64.6 | 628 | 30.0 | 3949 | 70.2 | 942 |
| | 8 | 65.6 | 234 | 74.8 | 345 | 92.9 | 272 | 88.6 | 944 | 91.3 | 231 | 69.9 | 628 | 33.3 | 3964 | 73.8 | 945 |
| ARM | 1 | 66.3 | 237 | 68.6 | 316 | 90.1 | 311 | 85.6 | 945 | 90.7 | 251 | 65.6 | 617 | 40.0 | 5413 | 72.4 | 1156 |
| | 8 | 67.2 | 234 | 69.6 | 322 | 93.9 | 306 | 93.1 | 933 | 93.3 | 242 | 71.8 | 623 | 40.0 | 5858 | 75.6 | 1217 |

## K   Further Discussion on Length-Penalty Strategies

### K.1   Implementations

To ensure fair comparisons, we follow the official settings of L1 [1] and THINKPRUNE [16], adopting their specified minimum allowed lengths when evaluating on easy tasks. We set the temperature to 0.6 and top-p to 0.95, consistent with both papers. Specifically, we use `L1-Qwen-1.5B-Exact`[5] at 512 tokens for L1 and `DeepSeek-R1-Distill-Qwen-1.5B-thinkprune-iter2k`[6] for THINKPRUNE.

### K.2   Inaccurate Estimation of Length-Penalty Strategies

We provide an in-depth analysis of L1 here, which applies a length penalty strategy during training. At inference time, users are required to explicitly specify token budgets in the instructions for the task. We follow the official setting and additionally extend the evaluation to the CSQA benchmark. Results are presented in Table 11. We clarify our claims as follows:

Users' inaccurate token estimation hurts performance, since the length penalty strategy assumes prior knowledge of the task to predefine the appropriate reasoning length. If the user's estimation is not accurate enough, the performance degradation is significant: *1)* **Underestimated budgets hurt performance on complex tasks.** On harder benchmarks like AIME, performance is severely degraded under low budgets: only 3.3% at 512 tokens, improving gradually to 26.7% at 3600 tokens. Such under-estimation of required reasoning length severely limits performance on challenging benchmarks. *2)* **Large budgets waste resources on simple tasks.** On easier benchmarks like CSQA, enforcing large reasoning budgets leads to token usage increases with minimal, or even detrimental, gains. For example, CSQA accuracy rises marginally from 45.8% at 512 tokens to 46.1% at 3600 tokens. In contrast, ARM learns to allocate longer reasoning only when necessary, avoiding both under- and overestimation.

### K.3   Further Comparison with L1

We further conduct a comparison between ARM-7B and `L1-Qwen-7B-Exact`[7]. For fairness, we align the backbone model by using DS-R1-Distill for both models. During L1 inference, we set token budgets to match ARM's average token usage on each dataset for fair comparison.

---

[5]https://huggingface.co/l3lab/L1-Qwen-1.5B-Exact
[6]https://huggingface.co/Shiyu-Lab/DeepSeek-R1-Distill-Qwen-1.5B-thinkprune-iter2k
[7]https://huggingface.co/l3lab/L1-Qwen-7B-Exact

As shown in Table 12, ARM consistently outperforms L1 with similar token usage, demonstrating the advantage of adaptive reasoning over length constraints. Notably, L1 requires human intervention to set the token budget manually, and an inaccurate assignment of the token budget would bring a performance drop (As detailed in Appendix K.2). In contrast, ARM autonomously adjusts its reasoning length based on task complexity through format selection, enabling better efficiency-performance trade-offs without manual token tuning.

## L  Limitations

**Dependency on Predefined Reasoning Formats**   In this work, we focus on four commonly used reasoning formats that generalize well across a wide range of reasoning tasks. However, we acknowledge that certain tasks may benefit from more specialized or nuanced reasoning strategies beyond this predefined set. Our reliance on predefined formats is primarily due to the limited capabilities of current models, which may struggle to autonomously identify or switch between diverse reasoning formats, let alone new reasoning formats. As a result, we define the formats in advance and introduce them through SFT to help the model establish a clear understanding of each reasoning type. We believe that as model capabilities continue to improve, future work can explore enabling models to autonomously select or even invent new reasoning formats without relying on predefined structures.

**Lack of Hard Task Data in Training**   Unlike some length-penalty-based strategies, our training setup does not include hard datasets such as prior AIME tasks, which may place our model at a disadvantage on hard tasks compared to methods like L1 [1] and THINKPRUNE [16] that incorporate such data. Nevertheless, ARM still shows clear improvements on base models and maintains stable performance on R1-distilled models on AIME'25, demonstrating its potential on hard tasks. We expect that incorporating harder data into training would further enhance performance. However, due to the high computational cost of reinforcement learning—and the current version of ARM being an early exploration aimed at evaluating generalization across tasks while improving token cost efficiency—we leave this extension to future work.

