# OpenReview forum: "ARM: Adaptive Reasoning Model"
_NeurIPS.cc/2025/Conference — NeurIPS 2025 spotlight_

### Official Review · Reviewer_PbWP · 2025-06-29

**Clarity:** 4
**Significance:** 4
**Originality:** 4
**Rating:** 6
**Confidence:** 2

**Summary:**

The authors propose an Adaptive Reasoning Model (ARM) that can modify how much reasoning is required based on the task difficulty. This ensures performance remains satisfactory without using unnecessary tokens by overthinking. It is trained in two stages: (1) Supervised Fine-tuning to learn reasoning formats, and (2) Reinforcement learning is then carried out to determine the ideal format selection given a task to avoid overthinking.

**Questions:**

I have no questions at this time.

**Ethical Concerns:**

["NO or VERY MINOR ethics concerns only"]

**Final Justification:**

Issues resolved:
Unclear where to refer to for "low-level technical implementation details"

Issues unresolved:
The paper avoids low-level technical implementation details by its nature and subject. It is a paper applying LLMs and adding on top of that some extra algorithms (e.g., Ada-GRPO) to essentially select the best way to reason based on the difficulty of the task.

I would say this is an excellent applied research paper, but it may not be a "timeless" paper (as it is based on modern-day GPT models, which are constantly being updated), and perhaps lacks significance in a theoretical research aspect. However, it could contribute as an intermediate step in researching better techniques for LLM reasoning. I think the expressed concerns on weaknesses among reviews is minor, and would suggest acceptance.

**Limitations:**

yes

**Paper Formatting Concerns:**

None to declare.

**Quality:**

4

**Strengths And Weaknesses:**

Strengths:

+ The paper is flawlessly written and a strong submission. Unfortunately, or not so unfortunately, I cannot articulate many comments as I have very little to critique. The authors motivate their research question extremely well, provide sufficient background, and refer to ample related work to argue the limitations of existing literature. Their work is beautifully presented, and the science is rigorous. Many metrics of interest are investigated and included here in the paper's results. Comparisons between different models are included as well.

Weaknesses:

- If I must provide a weakness, it is that the paper avoids low-level technical implementation details, but the methodology that the authors propose is explained very well.

---

> ### Author Rebuttal · Authors · 2025-07-31
>
> We thank the reviewer for the positive and encouraging feedback on the clarity, motivation, and rigor of our work. We welcome the opportunity to address your question below.
>
> ---
>
> ***[W1] Low-level technical implementation details.***
>
> [A1] Thanks for the comment. While the main paper focuses on explaining the methodology clearly, we also provide additional implementation details in the appendix for reproducibility. Appendix B details the dataset processing pipeline. Appendix D details training configurations for both the SFT and RL stages. Appendix G details the implementation of length-penalty baselines. We will provide more technical details in our next revision, and we are happy to provide further clarification to you if needed.

---

> > ### Comment · Reviewer_PbWP · 2025-08-03
> >
> > Thank you for your rebuttal and for referring me to the appropriate sections of the appendix. No further clarifications are needed

---

> > > ### Author Response · Authors · 2025-08-05
> > >
> > > Sincerely thank you for taking the time to review our paper. We are glad to know that our rebuttal addressed your concerns. Thanks for your engagement during the discussion!

---

### Official Review · Reviewer_iAYU · 2025-07-01

**Clarity:** 3
**Significance:** 2
**Originality:** 3
**Rating:** 5
**Confidence:** 3

**Summary:**

This paper proposes a framework to train an adaptive reasoning model. Specifically, the framework consists of two stages: (1) SFT to make the model familiar with four reasoning/answering modes. (2) Adaptive GRPO to reshape the reward for avoiding mode collapse when answering. The authors also conducted experiments across multiple reasoning benchmarks to demonstrate the effectiveness of their method in terms of token cost and accuracy.

**Questions:**

In my understanding, the reward shaping is based on the frequency of different modes. Is this implicitly conditioned on training data distribution? What if the correct answers of training data are biased to just one mode, and other modes will result in a low reward, will this still cause mode collapse? I'd like to see such an analysis, and if we have a way to mitigate this.

**Ethical Concerns:**

["NO or VERY MINOR ethics concerns only"]

**Final Justification:**

The authors' responses have resolved most of my concerns, and I appreciate the new results they are presenting.

**Limitations:**

See weakness.

**Quality:**

3

**Strengths And Weaknesses:**

**Strength**
- The method is reasonable and seems new to me (but I am not an expert in this field).
- The experiments show significant improvement over the baseline (GRPO).
- The paper is well-written and easy to follow.

**Weakness**
- (Major) The authors claimed in the related works that there exist other methods for efficient inference, which are based on "specialized and length-constrained model training". The authors also claimed that these estimations are "not always accurate". But I didn't see experimental support for this claim. An empirical comparison with these methods will make this work stronger.
- (Minor) This method requires manually defining four different reasoning modes (and their format/definition). It is not clear to me if (1) defining more fine-grained reasoning modes could help -> we have to rely on such definitions to achieve better reasoning performance (2) the four modes are enough for more open-ended questions beyond the benchmarks.

---

> ### Author Rebuttal · Authors · 2025-07-31
>
> We thank the reviewer for recognizing the novelty of our method, the strong experimental results, and the clarity of our writing. We appreciate the opportunity to address the concerns here.
>
> ---
>
> ***[W1] Empirical comparison with length-constrained methods.***
>
> [A1] Thanks for the suggestion. We provide an in-depth analysis of a previous wisdom method L1 [1] here, which applies a length penalty strategy during training. At inference time, users are required to explicitly specify token budgets in the instructions for the task. We follow the official setting [1] and additionally extend the evaluation to the CSQA benchmark. Results are presented in the table below. We clarify our claims as follows:
>
> Users’ inaccurate token estimation hurts performance, since the length penalty strategy assumes prior knowledge of the task to predefine the appropriate reasoning length. If the user’s estimation is not accurate enough, the performance degradation is significant: 1) **Underestimated budgets hurt performance on complex tasks.** On harder benchmarks like AIME, performance is severely degraded under low budgets: only 3.3% at 512 tokens, improving gradually to 26.7% at 3600 tokens. Such under-estimation of required reasoning length severely limits performance on challenging benchmarks. 2) **Large budgets waste resources on simple tasks.** On easier benchmarks like CSQA, enforcing large reasoning budgets leads to token usage increases with minimal, or even detrimental, gains. For example, CSQA accuracy rises marginally from 45.8% at 512 tokens to 46.1% at 3600 tokens. In contrast, ARM learns to allocate longer reasoning only when necessary, avoiding both under- and overestimation.
>
> | L1-Exact         | CSQA |       | AIME |       | MATH |       | AMC  |       | olympiad_bench |       |
> | ---------------- | ---- | ----- | ---- | ----- | ---- | ----- | ---- | ----- | -------------- | ----- |
> | specified budget | Acc. | #Tok. | Acc. | #Tok. | Acc. | #Tok. | Acc. | #Tok. | Acc.           | #Tok. |
> | 512              | 45.8 | 328   | 3.3  | 623   | 71.0 | 590   | 47.0 | 641   | 31.7           | 608   |
> | 1024             | 46.6 | 589   | 6.7  | 1291  | 77.2 | 1182  | 45.8 | 1283  | 37.2           | 1184  |
> | 2048             | 46.0 | 2004  | 13.3 | 1935  | 79.6 | 1751  | 55.4 | 1950  | 39.7           | 1813  |
> | 3600             | 46.1 | 4747  | 26.7 | 3696  | 81.8 | 3478  | 72.3 | 3525  | 43.7           | 3460  |
>
> ---
>
> ***[W2] Ablation study and generalization of reasoning formats.***
>
> [A2] Thanks for the thoughtful question. The four reasoning formats used in ARM are carefully designed to balance accuracy and token efficiency, and have proven effective across a variety of tasks. We also provide more detailed reasons in Section 3.1.
>
> To further investigate whether defining other reasoning formats would help, we add an additional reasoning format: function-calling, implemented via code execution. Specifically, during inference, when the model selects the Code reasoning format, we run the generated code using an interpreter to obtain an answer. If execution fails, the model falls back to simulating the output of the code [2].
>
> As shown in the table below, incorporating this function-calling format yields improvements, suggesting that more fine-grained formats can provide benefits. However, we also note that incorporating additional reasoning formats demands extra resources to implement the pipeline. For example, parallel function calls during training can lead to high memory consumption, and decreasing the number of processes may prolong the training time. Furthermore, formats like function-calling would introduce additional inference-time latency (e.g., due to runtime execution). Therefore, we adopt the current four reasoning formats for their widespread use and practicality, and leave the exploration of more reasoning formats to future work.
>
> | 7B Models              | CSQA | OBQA | GSM8K | MATH | SVAMP | BBH  | AIME'25 | Avg  |
> | ---------------------- | ---- | ---- | ----- | ---- | ----- | ---- | ------- | ---- |
> | vanilla ARM            | 86.1 | 84.4 | 89.2  | 73.9 | 92.0  | 61.4 | 16.7    | 72.0 |
> | ARM + function-calling | 86.1 | 84.5 | 90.3  | 74.3 | 92.8  | 62.1 | 16.7    | 72.4 |
> | Δ                      | +0.0 | +0.1 | +1.1  | +0.4 | +0.8  | +0.7 | +0.0    | +0.4 |
>
> We further examine the effect of removing specific reasoning formats on performance and token efficiency. Specifically, during inference, we remove the Direct Answer format on CSQA, Short CoT on GSM8K, and Long CoT on AIME’25. The results are summarized in the table below: On CSQA, removing the Direct Answer format increases token usage by +29.4% with negligible accuracy gain, showing it is crucial for efficiently handling simple tasks. In contrast, on AIME’25, removing Long CoT leads to a significant accuracy drop (-8.4), confirming its importance for complex reasoning. These results validate the necessity of the predefined reasoning formats in enabling adaptive reasoning.
>
> | 7B Models      | CSQA |        | GSM8K |        | AIME'25 |        |
> | -------------- | ---- | ------ | ----- | ------ | ------- | ------ |
> |                | Acc. | #Tok.  | Acc.  | #Tok.  | Acc.    | #Tok.  |
> | vanilla ARM    | 86.1 | 136    | 89.2  | 305    | 16.7    | 3253   |
> | after removing | 86.2 | 176    | 89.5  | 385    | 8.3     | 2137   |
> | ∆              | +0.1 | +29.4% | +0.3  | +26.2% | -8.4    | -34.3% |
>
> Our existing evaluation spans across in- and out-of-domain commonsense, mathematical, and symbolic reasoning. To further investigate the generalizability of ARM, we further extend the evaluation to two additional benchmarks: 1) GPQA, a complex QA dataset requiring compositional reasoning, and 2) StrategyQA, a multi-hop reasoning benchmark.
>
> As shown in the table below, ARM achieves comparable accuracy to the GRPO baseline while significantly reducing token cost by an average of 50%, and up to 65% in StrategyQA, consistent with our main findings. This demonstrates that ARM’s adaptive reasoning behavior generalizes well to more tasks. Thanks for your suggestion again, and we will update the results in our next revision.
>
> | 7B Models     | GPQA-Main |        | GPQA-Diamond |        | StrategyQA |        |
> | ------------- | --------- | ------ | ------------ | ------ | ---------- | ------ |
> |               | Acc.      | #Tok.  | Acc.         | #Tok.  | Acc.       | #Tok.  |
> | GRPO Baseline | 35.0      | 2324   | 37.4         | 2604   | 72.9       | 646    |
> | ARM           | 34.8      | 1306   | 36.9         | 1536   | 73.8       | 229    |
> | ∆             | -0.2      | -43.8% | -0.5         | -41.0% | +0.9       | -64.6% |
>
> ---
>
> ***[Q1] Will training data bias cause format collapse?***
>
> [A3] Thank you for the thoughtful question. You’re right, bias can indeed result from selecting unsuitable training data. To investigate this, we analyze an ARM trained solely on the AIME dataset (1983-2024), which primarily favors Long CoT solutions due to its competition-level complexity. We present the result in the table below. We evaluate this model on two simpler tasks—CSQA and OBQA—and observe that the model overwhelmingly selects Long CoT (~80%) even when simpler formats would suffice. This confirms that training on a biased dataset can indeed lead to over-reliance on a single reasoning format.
>
> |                         | CSQA     |               | OBQA     |               |
> | ----------------------- | -------- | ------------- | -------- | ------------- |
> |                         | Long CoT | Other Formats | Long CoT | Other Formats |
> | Training with AIME only | 79.4%    | 20.6%         | 83.0%    | 17.0%         |
>
> To mitigate this, ARM is trained on a diverse mixture of datasets across a wide range of difficulties. As shown in the table below, the full ARM recipe achieves both higher accuracy and significantly reduced token usage on the tasks, demonstrating that our approach effectively prevents format collapse and encourages adaptive reasoning behavior across domains.
>
> |                         | CSQA |       | OBQA |       |
> | ----------------------- | ---- | ----- | ---- | ----- |
> |                         | Acc. | #Tok. | Acc. | #Tok. |
> | Training with AIME only | 78.6 | 401   | 82.2 | 426   |
> | ARM Recipe              | 86.1 | 136   | 84.4 | 159   |
>
> ---
>
> **References**
>
> [1] Aggarwal P, Welleck S. L1: Controlling how long a reasoning model thinks with reinforcement learning[J]. arXiv preprint arXiv:2503.04697, 2025.
>
> [2] Li C, Liang J, Zeng A, et al. Chain of Code: Reasoning with a Language Model-Augmented Code Emulator[C]//International Conference on Machine Learning. PMLR, 2024: 28259-28277.
>
> ---
>
> Thank you once again for your feedback. We welcome any additional concerns or suggestions you may have.

---

> > ### Comment · Reviewer_iAYU · 2025-08-06
> > **Response to Author Rebuttal**
> >
> > Thanks for the detailed explanations and the additional experiments; they have resolved my concerns and have made the paper stronger. Please include them in the final version. I will consider raising my score to 5.

---

> > > ### Author Response · Authors · 2025-08-06
> > >
> > > Thank you for reading our response. We’re glad the rebuttal addressed your concerns and appreciate your insightful feedback. We’ll incorporate the discussion in the next revision. Please feel free to follow up with any further questions.

---

### Official Review · Reviewer_Ufiu · 2025-07-02

**Clarity:** 3
**Significance:** 3
**Originality:** 3
**Rating:** 5
**Confidence:** 4

**Summary:**

This paper introduces the Adaptive Reasoning Model (ARM), a framework designed to address the overthinking problem in large reasoning models.. ARM is trained to adaptively select from four distinct reasoning formats—Direct Answer, Short CoT, Code, and Long CoT—based on task difficulty. The training involves a two-stage process, SFT followed by a RL stage. For the RL stage, the authors propose Ada-GRPO, which incorporates a format diversity reward to prevent "format collapse" to a single, high-accuracy format. Experiments show that ARM achieves comparable accuracy to models using only Long CoT while reducing token usage by an average of 30% and speeding up training by approximately 2x.

**Questions:**

Questions
- How do you organize the SFT dataset? Does every question has 4 different types of answers?
- Is pass@1 actually avg@8 for multiple samples? This would effectively reduce randomness.

**Ethical Concerns:**

["NO or VERY MINOR ethics concerns only"]

**Final Justification:**

As in my initial review, this paper is technically solid and the experiments are sufficient. I suggest acceptance

**Limitations:**

Yes

**Quality:**

3

**Strengths And Weaknesses:**

Generally, I think this is a good work tackling the overthinking problem.
Strengths
- The experiments are sufficient to support the main claims, especially the additional experiments in Appendix
- The observation that ARM models learned to select reasoning formats based on task difficulty is convincing
- The flexibility of ARM, both enabling autonomous pattern selection and explicit guidance is a preferable feature.

Weakness
- The reported delta in Table 1 is maj@8, and the performance drop in pass@1 would be greater than that. This means ARM reduces generation length at the cost of performance degration.
- Baselines are not sufficient. Related works such as L1, ThinkPrune should be included in the main experiments for fair comparision.
- Experiments are only conducted on Qwen models. While recent works figure out potential issues of Qwen, it would be better to try more models like LLaMA and Mistral.

---

> ### Author Rebuttal · Authors · 2025-07-31
>
> We thank the reviewer for acknowledging the value of our proposed method and the supporting experiments. Below, we address your concerns in detail.
>
> ---
>
> ***[W1] Performance and efficiency trade-off balance.***
>
> [A1] Thank you for the comment. We acknowledge that Pass@1 may show a drop compared to Maj@8, as it does not fully reflect the model’s capacity to arrive at the correct answer. To mitigate this, we additionally adopt Maj@8, which helps reduce bias and variability associated with a single generation. It provides a more robust and reliable estimate of model performance in stochastic generation settings [1].
>
> Moreover, despite the performance drop under Pass@1, ARM achieves a better effectiveness-efficiency trade-off balance, reducing token usage while maintaining strong performance. This demonstrates ARM’s ability to reason adaptively, effectively balancing accuracy and cost.
>
> ---
>
> ***[W2] Comparison with length penalty baseline.***
>
> [A2] Thanks for the constructive feedback. In response, we conduct a comparison between ARM-7B and L1-Qwen-7B-Exact (Due to ThinkPrune only having a 1.5B model and time limits for reproducing it from scratch, we leave it to future study). For fairness, we align the backbone model by using DeepSeek-R1-Distill-Qwen-7B for both models. During L1 inference, we set token budgets to match ARM’s average token usage on each dataset for fair comparison.
>
> As shown in the table below, ARM consistently outperforms L1 with similar token usage, demonstrating the advantage of adaptive reasoning over length constraints. Notably, L1 requires human intervention to set the token budget manually, and an inaccurate assignment of the token budget would bring a performance drop (Please refer to our discussion on token budget estimation for length-constrained methods, presented in Response to Reviewer iAYU, A1, if you are interested). In contrast, ARM autonomously adjusts its reasoning length based on task complexity through format selection, enabling better efficiency-performance trade-offs without manual token tuning.
>
> |      |      | CSQA |       | OBQA |       | GSM8K |       | MATH |       | SVAMP |       | AIME'25 |       | BBH  |       | Avg  |       |
> | ---- | ---- | ---- | ----- | ---- | ----- | ----- | ----- | ---- | ----- | ----- | ----- | ------- | ----- | ---- | ----- | ---- | ----- |
> |      | k    | Acc. | #Tok. | Acc. | #Tok. | Acc.  | #Tok. | Acc. | #Tok. | Acc.  | #Tok. | Acc.    | #Tok. | Acc. | #Tok. | Acc. | #Tok. |
> | ARM  | 1    | 66.3 | 237   | 68.6 | 316   | 90.1  | 311   | 85.6 | 945   | 90.7  | 251   | 40.0    | 5413  | 65.6 | 617   | 72.4 | 1156  |
> |      | 8    | 67.2 | 234   | 69.6 | 322   | 93.9  | 306   | 93.1 | 933   | 93.3  | 242   | 40.0    | 5858  | 71.8 | 623   | 75.6 | 1217  |
> | L1   | 1    | 62.4 | 232   | 69.2 | 341   | 89.8  | 273   | 85.9 | 943   | 89.7  | 231   | 30.0    | 3949  | 64.6 | 628   | 70.2 | 942   |
> |      | 8    | 65.6 | 234   | 74.8 | 345   | 92.9  | 272   | 88.6 | 944   | 91.3  | 231   | 33.3    | 3964  | 69.9 | 628   | 73.8 | 945   |
>
> ---
>
> ***[W3] Experiment on LLaMA Baseline.***
>
> [A3] Thanks for the suggestion. Based on the superior performance of the Qwen model and its inherent adaptability to reinforcement learning [2, 3], we adopt Qwen as the backbone model in our experiments. We further explore models trained on LLaMA-3.2-3B and report the results below. Consistent with our main findings, ARM achieves comparable performance to the GRPO baseline while using fewer tokens across diverse task domains and complexity levels, demonstrating that our methods can generalize to other backbone models.
>
> We also note that the token reduction on LLaMA-based models is less pronounced than on Qwen (e.g., 15.7% reduction on GSM8K vs 55.2% on Qwen-base). Upon closer analysis, we find this is caused by repetitive outputs on occasion produced by the LLaMA-based model—a phenomenon also observed in prior work [3]—which may lead to longer response lengths. This discrepancy may stem from differences in model architecture or pretraining data, and we leave further investigation to future work.
>
> | LLaMA-3.2-3B  |      | CSQA |        | GSM8K |        | AIME'25 |        |
> | ------------- | ---- | ---- | ------ | ----- | ------ | ------- | ------ |
> |               | k    | Acc. | #Tok.  | Acc.  | #Tok.  | Acc.    | #Tok.  |
> | GRPO Baseline | 1    | 76.4 | 347    | 87.5  | 677    | 3.3     | 4616   |
> |               | 8    | 76.8 | 350    | 90.3  | 662    | 3.3     | 4375   |
> | ARM           | 1    | 76.2 | 158    | 86.1  | 546    | 3.3     | 3534   |
> |               | 8    | 76.5 | 162    | 89.8  | 558    | 3.3     | 3713   |
> | ∆             |      | -0.3 | -53.7% | -0.5  | -15.7% | 0       | -15.1% |
>
> ---
>
> ***[Q1] How to organize the SFT dataset? Does every question have 4 different types of answers?***
>
> [A4] Yes, every question has four different types of answers. Specifically, we collect seed question data from AQuA-Rat. While Direct Answer and Short CoT rationales are directly provided within the dataset, we augment it with Code and Long CoT rationales using GPT-4o and Deepseek-R1, respectively. Moreover, all rationales in the code reasoning format are verified for executability using an external Python interpreter. To ensure high-quality supervision, we filter out instances whose rationales yield incorrect answers, resulting in a curated dataset of 10.8K distinctive questions spanning all four reasoning formats. During SFT, the dataset is expanded to 10.8K * 4 = 43.2K training examples, which are shuffled and randomly fed into training, allowing the model to build a foundational understanding of different reasoning formats.
>
> ---
>
> ***[Q2] Clarification of evaluation metrics.***
>
> [A5] Thanks for the question. Pass@1 refers to the accuracy of the first sampled response. To quantify randomness, we rerun the evaluation multiple times and report the deviation across runs. As shown in the table below, the performance fluctuation is minimal across all benchmarks, typically within ±0.2 to ±0.5, and slightly higher (±1.6) only on AIME due to its small dataset size with only 30 instances and higher difficulty. These results suggest that the observed randomness is slight and within an acceptable range, supporting the reliability of our reported numbers.
>
> | CSQA | OBQA | GSM8K | MATH | SVAMP | AIME'25 | BBH  |
> | ---- | ---- | ----- | ---- | ----- | ------- | ---- |
> | ±0.4 | ±0.5 | ±0.2  | ±0.2 | ±0.3  | ±1.6    | ±0.5 |
>
> ---
>
> **References**
>
> [1] Zhang Q, Lyu F, Sun Z, et al. A Survey on Test-Time Scaling in Large Language Models: What, How, Where, and How Well?[J]. arXiv preprint arXiv:2503.24235, 2025.
>
> [2] Yang A, Yang B, Zhang B, et al. Qwen2. 5 Technical Report[J]. CoRR, 2024.
>
> [3] Wang Z, Zhou F, Li X, et al. Octothinker: Mid-training incentivizes reinforcement learning scaling[J]. arXiv preprint arXiv:2506.20512, 2025.
>
> ---
>
> If the reviewer has any further questions or requires clarification on any point, please feel free to ask. We are committed to making any necessary revisions to further improve our work.

---

> > ### Comment · Reviewer_Ufiu · 2025-08-06
> >
> > Thanks for the reponse. I will maintain my positive score

---

> > > ### Author Response · Authors · 2025-08-06
> > >
> > > Thank you for your engagement during the discussion. If you have any further questions, please feel free to follow up at any time.

---

### Official Review · Reviewer_XxN4 · 2025-07-03

**Clarity:** 3
**Significance:** 3
**Originality:** 2
**Rating:** 4
**Confidence:** 4

**Summary:**

The paper proposes Adaptive Reasoning Model (ARM) to reduce unnecessary long reasoning in LLMs by selecting more efficient formats without sacrificing accuracy. ARM supports four reasoning styles, Direct Answer, Short CoT, Code, and Long CoT, and uses a two-stage training process: supervised fine-tuning on multi-format data, followed by reinforcement learning with Ada-GRPO, a method that encourages diverse format use and prevents overuse of Long CoT. At inference, ARM can adaptively choose formats, follow user instructions, or use a consensus strategy. Experiments across multiple tasks and model sizes show that ARM achieves similar accuracy to baselines while cutting token usage by ~30% and halving RL training time.

**Questions:**

**1. Limited task diversity and generalization concerns**

- The evaluation and training setup are heavily concentrated on mathematical reasoning. All reinforcement learning datasets (GSM8K, MATH) and the supervised fine-tuning dataset (AQuA-Rat) focus on math problems, with only CSQA providing limited non-math signals.

- The evaluation includes only one hard benchmark (AIME’25), which also belongs to the math domain. Other difficult tasks, such as complex reasoning (e.g., GPQA, coding), are not considered. This raises concerns about the generalizability of ARM's adaptive behavior.

Could the authors evaluate ARM's performance on a broader range of challenging, non-mathematical tasks?

**2. Training and inference efficiency trade-offs**

- The ARM framework introduces a non-trivial training and data preparation burden. Stage 1 SFT requires each question to be annotated with four distinct rationales and validated for correctness, while Stage 2 RL involves about 1.4 M rollouts per model. This pipeline is more complex than alternatives like length-penalty fine-tuning.

- At inference time, while Adaptive Mode reduces token usage overall, the model still generates Long CoT for 10–20% of easy tasks (Fig. 2), indicating residual inefficiency. Additionally, Consensus-Guided Mode entails multiple generation passes (three efficient formats plus a potential fallback to Long CoT), which can substantially increase cost in high-reliability scenarios.

Could the authors consider potential simplifications of the training pipeline and quantify inference-time overhead under different usage settings?

**3. Lack of ablation on format set**
- ARM’s design relies on a fixed set of four reasoning formats, but the paper does not justify why these four were chosen or whether they are sufficient. Alternatives like diagram, or tool-calling scratch-pad, are not explored.

- No ablation tests whether removing or adding formats affects performance, token efficiency, or Ada-GRPO’s robustness against format collapse. While Section 4.4 compares formats during inference, it does not analyze the impact of modifying the format set during training.

An ablation study varying the training format set would strengthen the paper’s claims.

**4. Potential over-fitting**

- The RL stage uses CSQA, GSM8K and MATH, which also appear in evaluation. Even though train/test splits differ, these benchmarks are small and easily memorized.

How does ARM perform on completely unseen easy/medium/hard benchmarks (e.g. StrategyQA)?

**5. Need for additional experiments**

- Ada-GRPO introduces new hyper-parameters, such as the format-diversity scaling factor, but the authors have not assessed the sensitivity.

- Comparisons with baselines need to be more rigorous: GRPO is evaluated without token constraints, while length-penalty methods (e.g., L1, THINKPRUNE) are tested only on easy tasks. A stronger comparison would include GRPO combined with length penalties and report full token usage under maj@8 or Consensus Mode, rather than per-sample averages.

- ARM claims a 2× training speed-up, no absolute compute cost (e.g., wall-clock hours) is reported.

**Ethical Concerns:**

["NO or VERY MINOR ethics concerns only"]

**Final Justification:**

The authors’ responses have addressed most of my concerns, and I am maintaining my positive score.

**Limitations:**

Yes

**Quality:**

3

**Strengths And Weaknesses:**

**Strengths**

- Clarity: The paper is clearly written, with well-illustrated figures, concrete examples, and mostly reproducible implementation details.
- Empirical Results: Strong performance across a variety of reasoning benchmarks and model sizes, with significant token savings and faster training.
- Practical Impact: The adaptive reasoning mechanism is lightweight to implement and immediately useful for improving RLVR efficiency.

**Weaknesses**
- Limited Evaluation Scope: Hard tasks are restricted to math (AIME’25); generalization to non-math domains remains untested.
- Format Set is Fixed: No ablation on removing or adding reasoning formats; it is unclear whether the approach scales or generalizes beyond the current design.
- Training Cost: Requires curated multi-format annotations and extensive RL rollouts; this may hinder adoption in low-resource settings.
- Baseline Fairness: Limited comparisons with other efficiency methods; length-penalty baselines are not directly matched to ARM’s RL setup.
- Minor Clarity Gap: Some training hyperparameters (e.g., decay schedule) are only described in the appendix.

---

> ### Author Rebuttal · Authors · 2025-07-31
>
> We thank the reviewer for recognizing our paper’s clarity, strong empirical results, and practical impact. We appreciate the thoughtful and constructive comments and the opportunity to address the concerns raised.
>
> ------
>
> ***[W1, Q1, Q4] Limited task diversity and generalization concerns.***
>
> [A1] Thank you for the question. Our selection of evaluation testbed is based on both domain and complexity; thus, we incorporate the current seven sets to demonstrate the generalization, ranging from easy commonsense to symbolic and complex mathematical reasoning. We also acknowledge that GPQA and StrategyQA are two good choices to demonstrate the performance of ARM, and we extend our evaluation to these two additional benchmarks based on Qwen2.5-7B.
>
> As shown in the table below, ARM achieves comparable accuracy to the GRPO baseline while significantly reducing token cost by an average of 50%, and up to 65% in StrategyQA, consistent with our main findings. This demonstrates that ARM’s adaptive reasoning behavior generalizes well to more tasks. Thanks for your suggestion again, and we will update the results in our next revision.
>
> |7B Models|GPQA-Main||GPQA-Diamond||StrategyQA||
> |-|-|-|-|-|-|-|
> ||Acc.|#Tok.|Acc.|#Tok.|Acc.|#Tok.|
> |GRPO Baseline|35.0|2324|37.4|2604|72.9|646|
> |ARM|34.8|1306|36.9|1536|73.8|229|
> |∆|-0.2|-43.8%|-0.5|-41.0%|+0.9|-64.6%|
>
> ---
>
> ***[W2, Q3] Lack of ablation on format set.***
>
> [A2] Thanks for the thoughtful question. The four reasoning formats used in ARM are carefully designed to balance accuracy and token efficiency, and have proven effective across a variety of tasks. We also provide more detailed reasons in Section 3.1.
>
> To further investigate whether defining other reasoning formats would help, we add an additional reasoning format: function-calling, implemented via code execution. Specifically, during inference, when the model selects the Code reasoning format, we run the generated code using an interpreter to obtain an answer. If execution fails, the model falls back to simulating the output of the code [1].
>
> As shown in the table below, incorporating this function-calling format yields improvements, suggesting that more fine-grained formats can provide benefits. However, we also note that incorporating additional reasoning formats demands extra resources to implement the pipeline. For example, parallel function calls during training can lead to high memory consumption, and decreasing the number of processes may prolong the training time. Furthermore, formats like function-calling would introduce additional inference-time latency (e.g., due to runtime execution). Therefore, we adopt the current four reasoning formats for their widespread use and practicality, and leave the exploration of more reasoning formats to future work.
>
> |7B Models| CSQA | OBQA | GSM8K | MATH | SVAMP | BBH  | AIME'25 | Avg  |
> |-|-|-|-|-|-|-|-|-|
> |vanilla ARM|86.1|84.4|89.2|73.9|92.0|61.4|16.7|72.0|
> |ARM + function-calling|86.1|84.5|90.3|74.3|92.8|62.1|16.7|72.4|
> | Δ |+0.0|+0.1|+1.1|+0.4|+0.8|+0.7|+0.0|+0.4|
>
> We further examine the effect of removing specific reasoning formats on performance and token efficiency. Specifically, during inference, we remove the Direct Answer format on CSQA, Short CoT on GSM8K, and Long CoT on AIME’25. The results are summarized in the table below: On CSQA, removing the Direct Answer format increases token usage by +29.4% with negligible accuracy gain, showing it is crucial for efficiently handling simple tasks. In contrast, on AIME’25, removing Long CoT leads to a significant accuracy drop (-8.4), confirming its importance for complex reasoning. These results validate the necessity of the predefined reasoning formats in enabling adaptive reasoning.
>
> |7B Models|CSQA||GSM8K||AIME'25||
> |-|-|-|-|-|-|-|
> ||Acc.|#Tok.|Acc.|#Tok.|Acc.|#Tok.|
> |vanilla ARM|86.1|136|89.2|305|16.7|3253|
> |after removing|86.2|176|89.5|385|8.3|2137|
> |∆|+0.1|+29.4%|+0.3|+26.2%|-8.4|-34.3%|
>
>
> ---
>
> ***[W3, Q2, Q5] Training cost and inference overhead.***
>
> [A3] Thank you for the valuable comments. We acknowledge that preparing multi-format data introduces some annotation overhead. However, our SFT data pipeline is clear, lightweight, and practical: we leverage existing Direct Answer and Short CoT rationales in the original dataset, and construct the remaining formats (Code and Long CoT) using GPT-4o and DeepSeek-R1 via well-defined prompts (Appendix B). This process is automated, reproducible, and not resource-prohibitive, especially considering the benefit of enabling the model to build a foundational understanding of different reasoning formats.
>
> As for the RL stage, Ada-GRPO is based on GRPO, which is substantially more memory- and compute-efficient than other RL algorithms, like PPO [2]. GRPO is also commonly employed in length-penalty methods such as L1 and ThinkPrune. Our proposed Ada-GRPO further improves efficiency, reducing training time by half relative to vanilla GRPO through adaptive format selection, as discussed in Section 5.2. In terms of wall-clock training time, ARM-7B completes in 37 hours, making it ~2× faster than the GRPO baseline, which takes 64 hours.
>
> Regarding inference cost, although Long CoT is still invoked for easy tasks, ARM significantly reduces the token usage overall. Additionally, the three proposed reasoning modes are designed to support flexibility under different deployment requirements: 1) Adaptive Mode strikes a superior balance. 2) Instruction-Guided Mode offers a clear advantage when the assigned reasoning format is appropriate.3) Consensus-Guided Mode, while more costly, is performance-oriented and applicable in high-reliability settings. The inference token overhead associated with each mode is illustrated in Table 2 in Section 4.3. Below, we further report the inference time overhead under different mode settings, using Adaptive Mode as the baseline (normalized to 1) based on response length.
>
> ||Adaptive|Inst-Direct|Inst-Short CoT|Inst-Code|Inst-Long CoT|Consensus|
> |-|-|-|-|-|-|-|
> |Time|1.00|0.02|0.59|0.65|1.37|2.42|
>
> ---
>
> ***[W4, Q5] Comparison with length penalty baseline.***
>
> [A4] Thank you for the constructive feedback. In response, we conduct a comparison between ARM-7B and L1-Qwen-7B-Exact, which combines GRPO with a length penalty strategy. Due to the limited availability of ThinkPrune—which is only released for a 1.5B model—and time constraints preventing reproduction from scratch, we leave its comparison to future work. For fairness, we align the backbone model by using DeepSeek-R1-Distill-Qwen-7B for both models. During L1 inference, we set token budgets to match ARM’s average token usage on each dataset for fair comparison.
>
> As shown in the table below, ARM consistently outperforms L1 with similar token usage, demonstrating the advantage of adaptive reasoning over length constraints. Notably, L1 requires human intervention to set the token budget manually, and an inaccurate assignment of the token budget would bring a performance drop (Please refer to our discussion on token budget estimation for length-constrained methods, presented in Response to Reviewer iAYU, A1, if you are interested). In contrast, ARM autonomously adjusts its reasoning length based on task complexity through format selection, enabling better efficiency-performance trade-offs without manual token tuning.
>
> |||CSQA||OBQA||GSM8K||MATH||SVAMP||AIME'25||BBH||Avg||
> |-|-|-|-|-|-|-|-|-|-|-|-|-|-|-|-|-|-|
> ||k|Acc.|#Tok.|Acc.|#Tok.|Acc.|#Tok.|Acc.|#Tok.|Acc.|#Tok.|Acc.|#Tok.|Acc.|#Tok.|Acc.|#Tok.|
> |ARM|1|66.3|237|68.6|316|90.1|311|85.6|945|90.7|251|40.0|5413|65.6|617|72.4|1156|
> ||8|67.2|234|69.6|322|93.9|306|93.1|933|93.3|242|40.0|5858|71.8|623|75.6|1217|
> |L1|1|62.4|232|69.2|341|89.8|273|85.9|943|89.7|231|30.0|3949|64.6|628|70.2|942|
> ||8|65.6|234|74.8|345|92.9|272|88.6|944|91.3|231|33.3|3964|69.9|628|73.8|945|
>
>
> ---
>
> ***[W5, Q5] Some training hyperparameters of Ada-GRPO description.***
>
> [A5] Thank you for the thoughtful comment. We extend GRPO into Ada-GRPO by introducing two key components: 1) Format Diversity Scaling Factor to promote format exploration, and 2) Decay Factor to gradually shift focus to performance optimization.
>
> The scaling factor $G/F(o_i)$ is motivated by the group-wise nature of GRPO, where rewards are normalized within the groups. By shaping the reward in this way, Ada-GRPO effectively prevents format collapse into Long CoT, encouraging the model to explore diverse efficient reasoning formats. The scaling factor changes with the decay factor: as steps increase, the scaling factor decays from $G/F(o_i)$ to 1, thereby adjusting its sensitivity. This design helps prevent long-term misalignment due to the over-rewarding of rare formats, ensuring that the model transitions from exploration to exploitation in a controlled manner.
>
> We provide a systematic evaluation of this in Appendix A.2, showing that the decay schedule leads to smoother and more stable accuracy improvements, especially during mid-to-late training. We agree that more sensitivity analysis (e.g., $2*G/F(o_i)$) is valuable, and we plan to expand on this in future work.
>
> ---
>
> **References**
>
> [1] Li C, Liang J, Zeng A, et al. Chain of code: Reasoning with a language model-augmented code emulator[J]. arXiv preprint arXiv:2312.04474, 2023.
>
> [2] Shao Z, Wang P, Zhu Q, et al. Deepseekmath: Pushing the limits of mathematical reasoning in open language models[J]. arXiv preprint arXiv:2402.03300, 2024.
>
> ---
>
>
> Thank you for your thoughtful suggestions and careful review again. Please feel free to reach out if you have any additional concerns. We remain committed to refining and improving our paper.

---

> ### Comment · Reviewer_XxN4 · 2025-08-05
> **Official Comment by Reviewer XxN4**
>
> Thank you for your rebuttal and clarification. I have no further questions.

---

> > ### Author Response · Authors · 2025-08-06
> >
> > Thank you for reading our response and for your insightful feedback. Please feel free to follow up with any further questions. We remain committed to improving our paper.

---

### Note · Authors · 2025-08-13

Dear Reviewers, ACs, SACs, and PCs,

We sincerely appreciate the reviewers' feedback and encouraging recognition. We are encouraged by the consensus that our work offers:

1. **A well-motivated research question and novel method** for addressing overthinking in reasoning models through adaptive format selection (iAYU, PbWP).
2. **Strong empirical results** across diverse reasoning benchmarks and model sizes, demonstrating high performance with substantial token savings (XxN4, Ufiu, iAYU, PbWP).
3. **A lightweight and practical adaptive reasoning mechanism** that improves training efficiency and is easy to implement (XxN4, Ufiu).
4. **Clear and well-structured presentation** with illustrative figures, concrete examples, and reproducible implementation details (XxN4, iAYU, PbWP).

To further strengthen our position, we provide additional experiments and clarifications in our rebuttal:

1. **Generalization to more tasks**: We extend evaluation to GPQA and StrategyQA, showing that ARM maintains comparable accuracy to the GRPO baseline while reducing token usage by ~50% on average, and up to 65% in StrategyQA.

2. **Ablation study of reasoning formats**: Adding a function-calling format improves accuracy but incurs overhead, while removing formats reduces efficiency or accuracy, confirming the necessity of the current four formats.

3. **Comparison with length-penalty baseline**: With matched token usage, ARM outperforms L1-Exact, highlighting the advantage of adaptive reasoning over manual length constraints and avoiding the need for token budget tuning.

4. **Additional clarifications**: We provide further details on evaluation metrics, hyperparameters, and training data organization to enhance transparency and reproducibility.

We are glad our rebuttal addresses the raised concerns and believe these additions highlight our method’s effectiveness and generality. We will include the discussion in our next revision. We sincerely appreciate your time, effort, and thoughtful consideration.

---

### Decision · Program_Chairs · 2025-09-17

**Decision:**

Accept (spotlight)

**Comment:**

This paper proposes the Adaptive Reasoning Model (ARM), a framework to mitigate the overthinking problem in large reasoning models by dynamically selecting among four reasoning styles: Direct Answer, Short CoT, Code, and Long CoT. The model is trained in two stages: (1) supervised fine-tuning on multi-format data and (2) reinforcement learning with the proposed Ada-GRPO, which adds a format diversity reward to prevent mode collapse. At inference, ARM can adaptively select reasoning formats, follow user-specified preferences, or use a consensus strategy. Experiments on reasoning benchmarks, primarily with Qwen models, show that ARM maintains accuracy comparable to long CoT baselines while reducing token usage by ~30% and halving RL training time.

Strengths:
1. Ada-GRPO introduces a simple but effective mechanism to encourage diverse reasoning style usage and avoid collapse to long CoT.
2. Strong evidence across reasoning benchmarks that ARM can cut token usage while preserving accuracy, with faster training.
3. ARM provides a lightweight and implementable mechanism for efficiency that could be adopted in real-world RLVR pipelines.
4. ARM allows both adaptive selection of reasoning styles and explicit user control, which reviewers appreciated.

Weaknesses:
1. The approach relies on a manually defined set of four reasoning formats. It is unclear whether these are sufficient for broader tasks or whether more fine-grained or learned formats would help.
2. The reliance on curated multi-format annotations and extensive RL rollouts may hinder adoption in low-resource settings.
3. Some important hyperparameters and implementation details are only in the appendix, making reproduction less straightforward.

Overall, all the reviewers found the paper strong, clearly written, and addressing an important efficiency problem in reasoning LLMs.